

# Global Transition Rules for Translating Land-use Change (LUH2) To Land-cover Change for CMIP6 using GLM2

Lei Ma[1], George C. Hurtt[1], Louise P. Chini[1], Ritvik Sahajpal[1], Julia Pongratz[2], Steve Frolking[3], Elke Stehfest[4], Kees Klein Goldewijk[4,5], Donal O' Leary[1], Jonathan C. Doelman[4]

[1]Department of Geographical Sciences, University of Maryland, College Park, MD, USA

[2]Department of Geography, Ludwig-Maximilians-Universität, 80333 München, Germany and Max Planck Institute for Meteorology, Bundesstr. 53, 20143 Hamburg, Germany

[3]Institute of the Study of Earth Oceans and Space, University of New Hampshire, Durham, NH, USA

[4]PBL Netherlands Environmental Assessment Agency, The Hague, the Netherlands

[5]Copernicus Institute of Sustainable Development, Utrecht University, Utrecht, P.O. Box 80115, the Netherlands

*Correspondence to*: Lei Ma (lma6@umd.edu)

**Abstract.**

Anthropogenic land-use and land-cover change activities play a critical role in Earth system dynamics through significant

alterations to biogeophysical and biogeochemical properties at local to global scales. To accurately quantify the magnitude of these impacts, climate models need consistent land-cover and land-cover change time-series at a global scale, based on land-use information from observations or dedicated land-use change models. However, a specific land-use change cannot be unambiguously mapped to a specific land-cover change. Here, various transition rules are evaluated based on assumptions about the way land-use change could potentially impact land-cover. Building upon the latest Land Use Harmonization dataset

(LUH2), land-cover dynamics, particularly in forest cover and carbon stock, were simulated based on each rule from 850 to 2015 globally, at quarter degree spatial resolution. The resulting forest cover, carbon density, and carbon emissions for each rule were compared with those from remote sensing observations, U.N. Food and Agricultural Organization reports, and other studies. Examinations at global, country, and grid scales indicate that the optimal transition rule is for vegetation growing in primary and secondary land (including both forest and non-forest) to be completely cleared during the expansion of cropland,

urban land, and managed pasture, and to remain during rangeland expansion only if the land was originally non-forested. This confirms the transition rules suggested earlier in the HYDE dataset underlying LUH2. According to this rule, global forest area is estimated as 37.42 $10^6$ km$^2$, and forest area estimates at global and country scales both stay within the range derived from remote sensing products. This rule also mitigates the anomalously high carbon emissions observed in previous studies in the 1950s.





# 1 Introduction

Historical land-use activities have been significantly affecting the global carbon budget in both direct and indirect ways, and changing Earth's climate through altering land surface properties (e.g. surface albedo, surface aerodynamic roughness, and forest cover) (Betts, 2001;Bonan, 2008;Feddema et al., 2005;Guo and Gifford, 2002;Post and Kwon, 2000;Pongratz et al.,

2010;Brovkin et al., 2006;Claussen et al., 2001). During the past 300 years, >50% of the land surface has been affected by human land-use activities, >25% of forest has been permanently cleared, and 10-44 $10^6$ km² of land are recovering from previous human land-use disturbances (Hurtt et al., 2006). Impacts on the carbon cycle result from several processes: deforestation removes natural forest and its corresponding carbon biomass is used for wood products, burning, or decay by microbial decomposition (DeFries et al., 2002). Afforestation, in contrast, recovers forest which accumulates carbon but has a

lower maximum potential biomass than primary forest (Nilsson and Schopfhauser, 1995). Wood harvesting is one of the largest source contributing gross carbon emission by modifying the litter input into various soil pools, stand age, and biomass of secondary forest(Dewar, 1991;Nave et al., 2010;Hurtt et al., 2011). Cumulatively, land-use activities during 1870-2017 have contributed to a net flux of 190 Pg C carbon to the atmosphere (Houghton and Nassikas, 2017). While these emissions only account for 10% of current anthropogenic carbon emissions, they were a dominant contributor to increasing the atmospheric

$CO_2$ above pre-industrial levels before 1920 (Ciais et al., 2014).

Quantification of historical Land-Use and Land-Cover Change (LULCC) is important because it serves as the basis for examining the role of human activities in the global carbon budget and the resulting impacts to Earth's climate system. For this purpose, LULCC reconstructions enter Earth System Models (ESMs) (e.g., see (Lawrence et al., 2016) for land-use specific model simulations in the Coupled Model Intercomparison Project 6 (CMIP6), and (Brovkin et al., 2013) for CMIP5), Dynamic

Global Vegetation Models (DGVMs) (e.g., see (Le Quéré et al., 2018) for simulations with 16 DGVMs for the annual carbon budget estimates), and bookkeeping models (see (Le Quéré et al., 2018) for usage of the bookkeeping models by (Hansis et al., 2015) and (Houghton and Nassikas, 2017) to quantify the net land-use change carbon flux). Considerable efforts have been devoted to modelling historical land-use states (e.g. HYDE, SAGE) (Goldewijk et al., 2017;Kaplan et al., 2009) and land-use transitions (Hurtt et al., 2006;Hurtt et al., 2011;Houghton, 1999). In particular, the recent Land-Use Harmonization 2 (LUH2)

dataset(Hurtt et al., 2017b) harmonizes the most up-to-date historical data with 6 different future scenarios and provides global gridded land-use states and transitions in a consistent format for use in ESMs as part of CMIP6 experiments. However, large uncertainties still exist in the carbon/climate studies based on many of the above LULCC products (Houghton et al., 2012;Chini et al., 2012;Pongratz et al., 2014). For example, LULCC carbon emissions in CMIP5 have an anomalous spike during the years 1950-1960(Shevliakova et al., 2013). These anomalous emission estimates by ESMs (hereinafter referred to as the "pasture

anomaly") are caused by an implausible high conversion rate of natural and secondary vegetation to pasture, with the 1950s having double the conversion rate of the 40's or 60's (Shevliakova et al., 2013). Because of this, the simulated terrestrial land flux has a two decade delay in the switch from a land carbon source to a land carbon sink compared to observations (Shevliakova et al., 2013).



One reason for the above uncertainties is the lack of explicit global rules that translate land-use change estimates into land-cover changes, which is critical for ESM models (Di Vittorio et al., 2014;Di Vittorio et al., 2018;Brovkin et al., 2013;de Noblet-Ducoudré et al., 2012). Although land-use changes are generally associated with a change in land-cover and carbon stocks, these two changes are not always equivalent (see Figuere.1 in (Pongratz et al., 2018)),  and the degree of land-cover alteration

varies with the types of land-use changes. For example, the conversion from forested land to managed pasture and/or cropland tends to be associated with the full removal of native vegetation due to intensive human management, whereas vegetation may be less disturbed during the land conversion from non-forest (e.g. grassland) to rangeland. To enable the inclusion of such land-cover change processes, the HYDE 3.2 dataset has redefined the former pasture category used in CMIP5 into the two sub-categories of "managed pasture" and "rangeland" (with the total being termed "grazing land"). This redefinition intends

to suggest different treatments of vegetation and carbon removal in ESMs for these two types of land-use changes(Klein Goldewijk et al., 2017). However, explicit suggestions for land-cover and carbon stock modifications resulting from these new defined land-use types are not yet provided and the current split is based on an aridity index and population density (Klein Goldewijk et al., 2017) rather than actual information on underlying natural vegetation being transformed in their land-cover (e.g., clearing of forest for pasture) vs keeping their land-cover while being put under a different use (e.g., shrubland being

grazed without a transformation to a grassland). An inconsistent land-cover translation of these land-use products within an ESM or DGVM will potentially produce very different land-cover dynamics, which will impact the land surface biophysical and biochemical processes.

To reduce the uncertainties in estimating land-cover dynamics, this study investigates the impacts of land-use change on land-cover. Several alternative sets of transition rules are proposed and integrated into the Global Land use Model 2 (GLM2) model

(Hurtt et al., 2019, 2017b;Hurtt et al., 2017a) to simulate the forest cover and carbon dynamics. These simulations are then evaluated against estimates of contemporary forest cover and carbon density from remote sensing observations, and the resulting cumulative LULCC carbon emissions are compared with a range of other independent estimates. The goal is to propose an optimal transition rule for converting historical land-use changes (from LUH2) to land-cover changes for use in ESMs and DGVMs. This optimal rule combined with LUH2 could improve estimates of forest area and carbon stock at global,

country and grid-cell scales when compared to remote sensing data and reduce the 1950s pasture anomaly.

## 2 Methodology

In this study, two key land-cover properties (i.e. forest cover and vegetation carbon) are simulated by combining historical land-use change with transition rules. The historical land-use change information is specified by the LUH2 dataset (v2h, available at http://luh.umd.edu/) which serves as the forcing data for a new generation of advanced ESMs as part of CMIP6.

Section 2.1 describes the details of land-use change characterization, and section 2.2 defines each transition rule. The resulting forest cover and vegetation carbon is tracked at each grid cell (0.25×0.25˚) for the year 850 to 2015 using methods described in section 2.3 and 2.4. The simulated forest cover and vegetation carbon are then compared with multiple published datasets





of land-cover, including Global Land Cover Characterization (GLCC)(Loveland et al., 2000), Global Land Cover (GLC2000)(Bartholomé and Belward, 2005), GlobCover(Bicheron et al., 2008), the MODIS Land Cover Product(Friedl et al., 2010), forest cover products(DeFries et al., 2000;Hansen et al., 2010;FAO, 2015) carbon stock(Ruesch and Gibbs, 2008;Baccini et al., 2012), and estimates of land-use change emission(Pongratz et al., 2009;Houghton, 2010;Le Quéré et al., 2018;Stocker et al., 2011;Reick et al., 2010;Shevliakova et al., 2013;Houghton and Nassikas, 2017).

### 2.1 Land-use change characterization

LUH2 (version v2h) provides global, annual, gridded land-use states and transitions for the historical period 850-2015, and connects continuously to 6 different future scenarios from Integrated Assessment Models for the years 2015-2100 (Hurtt et al., 2017b). LUH2 accounts for diverse human-induced land-use activities including agricultural management, deforestation, and urbanization. LUH2 also includes bi-directional changes between natural forest and managed land (pasture and cropland) within a grid cell, including the effects of wood harvest and shifting cultivation. Since the rate of carbon loss due to deforestation is much faster than the carbon accumulation rate in the recovery process, using these gross land-use transitions helps to correct the underestimation of LULCC carbon emissions based only on the net transitions (Arneth et al., 2017).

The LUH2 dataset was generated with the GLM2 (Hurtt et al., 2017b;Hurtt et al., 2017a;Hurtt et al., 2019) which estimates annual sub-grid-cell land-use states and transitions using an accounting-based method. This model determines the fraction of every grid cell transitioning between each land-use type (e.g. primary land, cropland, urban) at each time step using multiple data-driven constraints including gridded patterns of historical land-use from the HYDE database(Goldewijk et al., 2017), historical national wood harvest reconstructions and potential biomass and recovery rates (Hurtt et al., 2006). Building upon previous work from CMIP5, for which the original LUH1 dataset was used, LUH2 has updated inputs from HYDE for historical agricultural patterns (Klein Goldewijk et al., 2017), a new historical wood harvest reconstruction, new maps and rates of shifting cultivation, extends the timespan to 850-2100 at 0.25×0.25˚, and constrains the forest cover gross transitions using remote sensing observations (Hansen et al., 2010). In addition, LUH2 includes 12 different land-use types (i.e. forested and non-forested primary and secondary land, cropland of C3 annual, C3 perennial, C4 annual, C4 perennial and C3 nitrogen-fixing, urban, managed pasture and rangeland) and includes transitions between all combinations of these categories.

In LUH2, "primary" refers to land previously undisturbed by any human activities, while "secondary" refers to land undergoing a transition or recovering from previous human activities. Global secondary land area was specified as zero in 850. Note that primary and secondary lands are further sub-divided into forested and non-forested grids using a definition based on the potential aboveground biomass density (forested land requiring an aboveground biomass density $\geq 2$ kg C/m$^2$).

### 2.2 Transition rules

Nine transition rules are proposed (Table 1) to analyse the effects of land-use change on land-cover dynamics, whereby each rule differs in treatment of vegetation cover and vegetation carbon stock during land-use changes. Rules 1-4 all assume complete clearance of vegetation for cropland and vary on vegetation clearance for managed pasture and rangeland. The rules





5-9 are added for analytical purposes, rather than as realistic possibilities. For example, rule 3 presumes all land-use changes alter land-cover and reduce carbon stock, and this rule would produce the least global forest cover and carbon stock. Rule 1 and 2 differ in treatment of vegetation in forested land when converted to managed pasture, and the resulting difference between their forest and carbon stocks indicate the impact of managed pasture expansion on forests, and also tests whether the

disaggregation of grazing land into managed pasture and rangeland will address the pasture anomaly issue in 1950-1960. Rule 1 (clearance of all vegetation for cropland and managed pasture, and only forest clearance for rangeland) is in fact the rule suggested in the underlying HYDE dataset and its distinction between pasture and rangeland (Klein Goldewijk et al 2017). For simplicity, we do not consider partial removal of vegetation in this study; vegetation is either fully removed or fully remains as these land-cover transitions represent the maximum and minimum bounds for land-cover alteration. In this study, the

transition rules are applied to all regions and are constant across the whole simulation period. Although the impacts of land-use change on land-cover may vary in different regions, the discussion of region-varied and time-varied transition rules is beyond the scope of this study.

It is important to note that these nine rules are not equally realistic, and the purpose of including some rules (labelled as analytical rules) is to investigate individual or joint contributions of cropland, managed pasture and rangeland expansion on

forest and carbon. For example, forest and carbon dynamic resulting from analytical rule 6 could suggest individual impact of cropland expansion.

**2.3 Simulation of land-cover change**

In this study, land-cover change is simulated within the GLM2 by combining land-use transition rates from LUH2 with each transition rule (Table 1) to track forest cover change and carbon dynamics at 0.25º spatial resolution. GLM2, a global extension

of the Miami ecosystem model(Lieth, 1975) is used to estimate the historical potential distribution of vegetation carbon stocks and carbon recovery rates of primary natural vegetation. The Miami model was run globally at 0.25×0.25˚ resolution using MSTMIP climatology(Wei et al., 2014), environmental factors were not taken into consideration such as $CO_2$ fertilization or nitrogen limitation. It resulted in an estimated global vegetation carbon stock (including above- and belowground) of 718 Pg C, and the resulting potential biomass map is shown in Figure 1a. For comparison, global potential vegetation carbon stock

was estimated as 557 Pg C in (Kucharik et al., 2000), 772 Pg C in (Pan et al., 2013) and 923 Pg C in (Sitch et al., 2003). Forested land in GLM2 is defined as land which has aboveground potential biomass of at least 2 Kg $C/m^2$(Hurtt et al., 2011). With this definition, global potential forest area was estimated as 47.82 million $km^2$, and the resulting potential forest cover map is shown in Figure 1b. For comparison, global potential forest area was estimated as 48.68 million $km^2$ in (Pongratz et al., 2008), and potential forests and woodlands area was 55.3 million $km^2$ in (Ramankutty and Foley, 1999).

When land is converted to cropland, managed pasture, and/or rangeland, each transition rule indicates that vegetation in primary and secondary may be cleared or remain intact as the result of land-use changes. For example, for a given land-use transition rate from forest to pasture, if the applied transition rule indicates to clear the vegetation completely, then the resulting grid cell vegetation fraction in forest land-use type is reduced equal to the amount of pasture gained. If the rule indicates not





to clear vegetation, then only the land-use type will be changed to pasture and the vegetation area will be unchanged, but the vegetation will be influenced by the management in terms of stand age/biomass, which are assumed to cease growing due to pressure from subsequent human management. If this pasture land is further converted to other non-primary and non-secondary land (e.g. cropland, rangeland or urban), the vegetation remaining from previous forest-pasture conversion then will be totally

cleared. Therefore, the vegetation fraction existing within the cropland, managed pasture, rangeland and urban of each grid-cell can be tracked via the following equation:

$$f(i, t+1) = f(i, t) + f^{gained}(i, t) - f^{lost}(i, t), (i = 5,6,7,8), \tag{1}$$

Where $f(i, t)$ is the fraction of grid-cell that is vegetated in land-use type $i$ (i.e. classes 5-8: cropland, managed pasture, rangeland, urban) at time $t$, $f^{gained}(i, t)$ and $f^{lost}(i, t)$ are the vegetation fractions gained or lost to/from land-use type $i$, and

they could be calculated:

$$f^{gained}(i, t) = \sum_{j=1}^{4} a_{ij} \gamma_{ij}, (i = 5,6,7,8; j = 1,2,3,4), \tag{2}$$

$$f^{lost}(i, t) = \frac{f(i,t)}{l(i,t)} \sum_{k=1, k \neq i}^{8} a_{ki}, (i = 5,6,7,8; k = 1,2,\cdots,8), \tag{3}$$

The possible values of $i$, $j$ and $k$ are 1, 2, … , 8 representing primary forested land, primary non-forested land, secondary forested land, secondary non-forested land, cropland, managed pasture, rangeland and urban respectively. $a_{ij}$ is the land-use

transition fraction estimate by LUH2 from land-use type $j$ (i.e. primary forested land, primary non-forested land, secondary forested land, secondary non-forested land) to land-use type $i$, $\gamma_{ij}$ represents the translator factor to convert land-use change to land-cover change, it equals to 1 if the transition rule in Table 1 indicates an 'X' or 'F' for this land-use change. For example, $\gamma_{ij}$ is 1 for land-use change from primary land (forested, non-forested grids) to cropland in rules 1 and 2, but 0 for the same type of change in rules 8 and 9. This translator factor is 1 for all types of land-use change in rule 3 since all vegetation is

cleared during all land-use changes. $l(i, t)$ is the fraction of land-use type $i$ at time $t$, and this fraction is larger than its vegetation fraction.

Vegetation in primary and secondary land can be recovered through the process of abandonment of these non-primary and non-secondary land-use. Note that reforestation but not afforestation is also considered in this study. The former is to re-establish forest on the land which has been forested before, while the latter is an anthropogenic activity to establish forests on land which has never been forested. Thus, the vegetation of primary and secondary land is tracked by the following equation:

$$f(i, t+1) = f(i, t) - f^{lost}(i, t) + f^{gained}(i, t), (i = 1,2,3,4), \tag{4}$$

$$f^{lost}(i, t) = \sum_{j=5}^{8} a_{ji} \gamma_{ji}, (i = 1,2,3,4; j = 5,6,7,8), \tag{5}$$

$$f^{gained}(i, t) = \sum_{k=5, k \neq i}^{8} \frac{f(k,t)}{l(k,t)} a_{ik}, (i = 1,2,3,4; k = 5,6,7,8), \tag{6}$$



Where $f(i,t)$ is fraction of vegetation at land-use category $i$ (primary forested land, primary non-forested land, secondary forested land, secondary non-forested land) at time $t$. $a_{ji}$ is land-use transition fraction from primary and secondary land to cropland, managed pasture, rangeland and urban in LUH2, $\gamma_{ji}$ is the translator factor, as is $\gamma_{ij}$ in Eq.2; both indicate whether to clear the vegetation during land-use changes. $f(k,t)$ and $l(k,t)$ are vegetation fraction and land-use fraction in land-use type $k$ (i.e. cropland, managed pasture, rangeland, urban), and $a_{ik}$ is land-use transition due to land-use abandonment. Therefore, the forest cover at time $t$ in these nine rules includes the vegetation originally growing in primary and secondary forested land, vegetation recovered from abandoned cropland, managed pasture and rangeland, and vegetation remaining in cropland, managed pasture, rangeland and urban which is not cleared during land-use change.

**2.4 Simulation of vegetation carbon dynamics**

Forest carbon stocks fluctuate through releasing and accumulating carbon in response to natural growing conditions, disturbances, and anthropogenic land-use changes, which can vary widely in terms of their carbon impacts. For land-use changes associated with clearing or harvesting vegetation, the forest biomass is either released immediately (e.g. burning) or stored in soil pools or as timber products (both of which eventually decay over decades). However, when managed land is abandoned and allowed to recover, the vegetation takes up $CO_2$ from the atmosphere through photosynthesis, resulting in increasing carbon stocks in vegetation and possibly soils. The magnitude of each of these bi-directional carbon flows ultimately determine if the land is a net carbon sink or carbon source. In this study, the temporal dynamics of carbon fluxes after land-use change are simplified, with all biomass (above- and below-ground) being released instantaneously to the atmosphere. Note that the biomass stock change is a rough proxy of actual net land-use change fluxes, for which delayed emissions from litter and soil carbon and product pools needed to be accounted for as well as instantaneous emissions from burning biomass. (Erb et al., 2018) noted that changes in soil carbon associated with loss of vegetation biomass are usually associated with carbon losses, but are likely less important than biomass changes, as are net fluxes from product pool changes.

Similar to land-cover change simulation in section 2.3, if transition rules indicate vegetation clearing at expansion of cropland, managed pasture, rangeland or urban land, vegetation biomass is totally released as a carbon emission, and its mean age is set as zero. If vegetation is not cleared based on transition rules, the biomass remains but ceases to increase, and the mean age of this vegetation also remains unaffected, because the mean age is used in this model only for the calculation of biomass density. Keeping age fixed corresponds to keeping biomass from further growing, which represents the influences of management. If the land is abandoned and converted back to secondary land, the biomass regrows towards equilibrium, and the mean age of vegetation increases year by year. Thus, the biomass density in secondary vegetation is calculated for each grid cell using its stand age, potential biomass, and potential NPP:

$$B(t) = B_0(1 - e^{-NPP_0 \times G(t)/B_0}) \ , \tag{7}$$

Where $B(t)$ is the aboveground biomass density of vegetation at secondary land at time $t$, and $B_0$ is the potential aboveground biomass density from Miami model and varied by grid location (shown in Figure.1a), and $NPP_0$ is the potential NPP of the



wood fraction, and $G(t)$ is the mean age of secondary vegetation. Note that $B_0$ and $NPP_0$ is constant over simulation period from 850 to 2015. Above- to below-ground biomass ratio is assumed as 3:1 when converting aboveground biomass to total biomass (above- and belowground), and biomass density is converted to carbon by a ratio of 0.5.

Plants cultivated by human management (e.g. crops and orchards) are not tracked in this study; zero biomass is assigned to
cropland, managed pasture, rangeland and urban use types. However, carbon is tracked for vegetation remaining from primary or secondary due to the land-cover transition rules, as well as lands that convert from human management back to natural lands. Thus, the total carbon stocks in this study should be lower than other estimates (Houghton, 2003;Saatchi et al., 2011), especially in the grids with a higher fraction of non-primary and non-secondary land-use.

## 2.5 Diagnostics for evaluating transition rules

To evaluate which transition rules best translate land-use changes to land-cover changes, the simulation results were compared with contemporary forest cover and carbon density maps from remote sensing observations and other estimates, as well as LULCC carbon emissions from other studies using different models. Contemporary values of forest cover and carbon density are used for two reasons. First is the lack of multiple diagnostics of forest cover and carbon density across the whole simulation period (i.e. 850 to 2015). Second is that contemporary values could potentially reflect cumulative error in converting land-use
change to land-cover change since 850. We assume that if a transition rule produces a best match with the diagnostic maps of forest cover and carbon density, then it would also produce the best estimate for the historical period.

To produce a reference map of contemporary forest cover, six widely used satellite-based land-cover and tree coverage datasets (Loveland et al., 2000;Bartholomé and Belward, 2005;Bicheron et al., 2008;Friedl et al., 2010;DeFries et al., 2000;Hansen et al., 2010) (see Table 2) are collected as well as the Global Forest Resources Assessment (FRA) 2015 (FAO, 2015). In Table
2, GLC, GLC2000, GlobCover and MODIS LC are land-cover datasets rather than tree cover and were produced based on different classification schemes resulting in different land-cover legends. Prior to being used as diagnostics in this study, they needed further reclassification of their land-cover legends into a common representation of forest canopy cover at the same spatial resolution (0.25°) by the following procedures: First, the GLCC, GLC2000, GlobCover and MODIS LC were converted to tree cover fraction based on Table S1 at their native resolutions (Song et al., 2014). Then, all six datasets were resampled to
1 km resolution and translated to a binary (forest versus non-forest) map by applying a 30% tree-cover threshold (Sexton et al., 2016). Through counting the percentage of pixels marked as forest within each 0.25x0.25° grid cell, six global gridded forest cover maps at 0.25º spatial resolution were generated. As these satellite-based datasets were developed from different sensors (e.g. AVHRR, SPOT-4, MERIS, MODIS, Landsat) and models (regression trees, decision tree, clustering labels and random forests), an averaged map (hereinafter referred to as 'Averaged satellite-based forest cover') was generated to examine
spatial pattern of contemporary forest cover simulated by each transition rule. In addition, since FAO only reports national forest cover (not spatially explicit), these data were only used for comparison at the country level.

Carbon density maps are employed as the second metric to evaluate the transition rules. Two datasets were employed: the IPCC Tier-1 biomass carbon map for the year 2000 (Ruesch and Gibbs, 2008) and a pantropical biomass map (hereinafter





referred to as the Baccini product (Baccini et al., 2012). The former, a global above- and below-ground carbon density map, is created by dividing the globe into 124 carbon zones by land-cover, continental regions, eco-floristic zones, and forest age and assigning each zone a unique carbon stock value. The latter is estimated by combining ground plots, GLAS LiDAR observations and optical reflectance of MODIS. This dataset employs the empirical relationship between aboveground biomass

and tree diameter at breast height and estimates aboveground biomass density for pantropical regions (40˚S-30˚N). Both carbon density maps were resampled to 0.25˚ before evaluation.

In addition, the ability of the transition rules to reproduce LULCC carbon emissions is also assessed. The estimates of LULCC carbon emissions were compiled from published papers (Table 3) (Reick et al., 2010;Stocker et al., 2011;Houghton, 2010;Houghton and Nassikas, 2017;Shevliakova et al., 2009;Pongratz et al., 2009;Le Quéré et al., 2018). These studies have

significant discrepancy in emissions estimates as they employed various methods (e.g. book-keeping methods and different process-based models), LULCC datasets, and considered different types of land-use change activities. They also differ in treatment of environmental change, for example, (Reick et al., 2010;Stocker et al., 2011;Shevliakova et al., 2009;Pongratz et al., 2009) include effects of evolving climate or atmospheric CO2 concentration on LULCC emissions, which is not accounted for in bookkeeping mode based studies(Houghton, 2010;Houghton and Nassikas, 2017). In this study, only the range of these

estimates during the pre-industrial and industrial periods are chosen to evaluate the transition rules. We posit that the optimal transition rule should not produce anomalous carbon emissions that are outside the compiled range.

In summary, the GLM2-based estimates of forest cover and carbon density in the year 2000 and LULCC carbon emissions during the periods 850-1850 and 1850-2000, based on nine different transition rules are compared with the above three types of diagnostics. The final determined optimal transition rules should produce: 1) the most accurate forest cover that has the

smallest difference with diagnostic maps at global, country and grid scale, and the total forest cover at global and country level should be within the range of diagnostics; 2) the closest carbon density map compared to diagnostics with the smallest difference and total carbon stock as well; and 3) reasonable LULCC carbon emissions within the range from other diagnostic estimates and minimizing the anomalous emissions during 1950-1960. Finally, if several rules have a reasonably good fit to these three diagnostics, other criteria, such as the definition characteristics for managed pasture and rangeland has handled in

HYDE(Klein Goldewijk et al., 2017) will also be taken into account in identifying the optimal rule.

# 3 Results

## 3.1 Forest cover evaluation

The global gridded forest cover maps resulting from rules 1-4 in 2000 are generally consistent in forest extent with satellite-based observations (shown in Figure 2). For example, they all estimate higher forest cover in tropical rainforest and northern

boreal forests but lower cover in western USA, eastern Australia, Eastern Europe and Central Asia. As rules 1, 2, and 3 only differ in whether to clear vegetation and carbon in the conversion from non-forest to pasture or rangeland, the forest cover resulting from rules 1, 2, and 3 are the same.





The total area of global forest in 850 amounts to 47.82 million km$^2$ according to the Miami model (Figure 1b and Figure 3a) when all forested lands were in a primary state by definition and decreased thereafter (Figure 3a). Forest loss has accelerated since the beginning of the Industrial Revolution and shows relatively high annual change rates (shown in Figure 3c). The transition rules produce a wide range of global forest cover in 2000 from 37.42 to 45.89 million km$^2$. In rules 1, 2, and 3, the global forest is lost at the highest rate due to all land-use change activities on forested land resulting in the clearing of forest, and only 37.42 million km$^2$ of global forest is left in 2000 under these three rules. In contrast, under rule 4 forest remains during conversion to rangeland expansion, and this would result in greater forest cover (e.g. 41.80 million km$^2$ in 2000, Table 4).

Six satellite-based forest cover datasets and FAO data report the global forest area around the year 2000 ranging from 35.79 to 42.74 million km$^2$. One of major reason underlying the discrepancy in global forest area is the difference in defining 'forest', particularly in the regions with intermediate tree cover (Sexton et al., 2016). The global forest area in the year 2000 resulting from our transition rules are compared to the range of seven diagnostic estimates (Figure 3b). The forest cover based on analytical rules 7, 8 and 9 is beyond the range of the diagnostics, indicating that these rules underestimate the impacts of land-use change on land-cover and overestimate the global forest existing in the present day. The excessive remaining forest cover in these three rules also rejects these rules' assumptions that only a particular type of land-use change would alter the land-cover. In contrast, rules1-4 produced estimates of global forest area within the range of diagnostics.

The forest cover estimation from transition rules are further compared with diagnostic datasets at the country level. In the diagnostic forest cover datasets, three-fourths of global forest cover lies within eight countries: the Russian Federation, Brazil, Canada, United of States of America, China, Democratic Republic of the Congo, Indonesia and Peru. Rules 1-4 also produce the same pattern of locating most forest land within these eight countries (Table 4). The forest cover estimates from rules 1-4 are generally well within the range of diagnostics for most of the eight countries (e.g. Brazil, Indonesia, and United States of America) in terms of forest area and slightly overestimated in the Russian Federation and Canada, where the estimates of rules 1-3 are closer to the upper bound of the diagnostics than rule 4.

These comparisons evaluate the accuracy of the transition rules in translating land-use change to land-cover change in terms of gross forest cover at global and country level. Further examination at the grid level is also needed. Since the FAO report only provides national forest cover, the averaged satellite-based forest cover map was used to calculate the average of absolute difference across global grids (Figure 4). Rules 1, 2, and 3 produce the smallest overall difference (i.e. below 90 km$^2$) with the averaged satellite-based forest cover map.

## 3.2 Evaluation of carbon dynamics

The net carbon emissions of the nine transition rules was calculated over two periods (850 to 1850 and 1850 to 2000) and compared to other studies (Table 5). Rules 1-4 produced similar patterns to other studies, specifically that global carbon emissions of 1850-2000 are twice as large as that of 850-1850. However, the emissions estimates of each period varied among



rules 1-4, from 55 to 77 Pg C during 850-1850 and from 142 to 185 Pg C during 1850-2000, due to the assumptions for clearing vegetation during land-use change. For example, rule 3 produced the largest emissions as the carbon in both forested and non-forested land is released for all land-use changes, and rule 1 produces fewer emissions since the vegetation is not cleared and carbon is not released when non-forested land is converted to rangeland. In general, rules 1, 2, 3 and 4 estimated comparable

emissions with other studies, while the emissions of the analytical rules 6-9 are out of range (Table 5).

Carbon emissions from pasture expansion were calculated for LUH1 (Hurtt et al., 2011) and this is used as a baseline to assess the improvement of transition rules on the pasture anomaly. Rules 1-4 estimate fewer emissions during this decade and decrease the anomaly between 4 to 10 Pg C. In LUH1, the anomalous emissions spike during 1950-1960 mainly arises from overestimating the emissions from pasture expansion, especially in four regions and countries (i.e. west and central Africa,

China, former USSR and South America excluding Brazil). The carbon flux from expansion of managed pasture and rangeland in LUH2 was significantly reduced at global (Figure 5) and regional (Figure 6) scales in simulations based on rules 1, 2, and 3. Note that the pasture land in LUH1 corresponds to rangeland and managed pasture together in LUH2. Rule 2 reduces the anomalous emissions more significantly than rule 1 (reduced 6 Pg C in rule 1 and 7 Pg C in rule 2), because rule 1 completely clears vegetation when transitioning to managed pasture, whereas rule 2 only removes vegetation if the preceding land cover

is primary or secondary forest.

Rules 1-4 generally capture the spatial pattern that carbon density in tropical rainforest regions is much higher than northern boreal forests (Figure 7). To further examine the spatial pattern of estimated carbon density, the estimates from all rules were compared to the carbon density maps of IPCC Tier 1 (above- and belowground) globally and the Bacchini' dataset (only aboveground) at the pantropical scale by calculating averaged absolute difference (Figure 8). According to this comparison,

rules 1 and 2 still best capture the carbon density heterogeneity with the bias less than 2.2 Kg C/m$^2$ at global comparison and produce bias less than 2 Kg C/m$^2$ for aboveground biomass at pantropical comparison.

The total carbon stock, grouped by forest cover using the averaged satellite-based forest cover map, from rules 1, 2, and 3 are compared with IPCC Tier1 and the Baccini product (Figure 9). Rules 1 and 2 still produce the closet carbon stock compared to the two diagnostic datasets, especially for grids with higher forest fraction (e.g. >50%), and slightly underestimate for grids

with higher fraction of non-forest land-use which may result from zero biomass assigned to these lands after land-use change.

## 4 Discussion and Conclusions

This study discussed possible alterations of land-cover as a result of prescribed land-use change and simulated the resulting forest cover and carbon dynamics through GLM2 model. The comparisons on forest cover, carbon stock and LULCC emissions ultimately indicates that both rules 1 and 2 could accurately translate land-use change to land-cover change and reproduce the

majority of current forest cover and plant carbon stock. Specifically, these rules state that the vegetation growing in primary and secondary land (both forested and non-forested) is completely cleared and all carbon released during the expansion of cropland and urban land, but vegetation remains only during rangeland expansion on non-forested land (rule 1) or remains





during managed pasture and rangeland expansion if the land is non-forested originally (rule 2). The vegetation remaining in managed pasture and rangeland is cleared when the land is subsequently converted to non-primary and non-secondary land (i.e. cropland, urban, managed pasture and rangeland). As a result, based on rule 1 (2), forest area decreased to 37.42 million km$^2$ in 2000, LULCC results in 70 (72) Pg C carbon emissions during 850-1850 and 170 (175) Pg C during 1850-2000, further

reducing pasture anomaly emissions by 6 (7) Pg C in 1950-1960s.

A key feature of this study is to explicitly link land-use change and land-cover change and to suggest a suitable method to incorporate the LUH2 land-use transition dataset into ESMs and DGVMs. The information from this study could facilitate reconstruction of historical land-cover change; building upon LUH2, the suggested transition rules could reproduce the smallest difference in contemporary forest cover and carbon stock with independent estimates from remote sensing. Currently,

rule 1 is recommended by LUH2 to translate the land-use change transitions into land-cover transitions in ESMs or DGVMs. While transition rule 2 generates a global forested area which is closer to the averaged remote sensing-based estimates, the difference in the forested area simulated by rules 1 and 2 is within the margin of uncertainty for remote sensing-based products, and is therefore scientifically insignificant. Therefore, recommendation of rule 1 over rule 2 is based on an assumption about the way in which rangeland versus managed pasture is established and managed, which is also consistent with the

recommendation in HYDE 3.2 dataset(Klein Goldewijk et al., 2017) that removes all vegetation when establishing cropland, urban land, or managed pasture, and leaves all vegetation when establishing rangeland, regardless of the underlying vegetation type.

More rigorous evaluation of the land-cover dynamics resulting from various transition rules from 850 to present is difficult because the available diagnostic datasets only document the land-cover and carbon stock in recent decades. For example, most

global satellite-based observations only estimate land-cover after 1980 (DeFries et al., 2000;Loveland et al., 2000;Bartholomé and Belward, 2005;Bicheron et al., 2008;Friedl et al., 2010;Hansen et al., 2010). Alternatively, contemporary measurements were used as diagnostics to assess the translation accuracy of each transition rule. This is because, in principle, the effects of prior land-use change activities before 2000 are manifested in the current state of land-cover (e.g. forest cover and carbon stock). As the current land-cover state is the cumulative sum of natural state and alterations from previous land-use change,

the error of incorrect translation of land-use change to alterations on land-cover will also be accumulated throughout and eventually result in a biased estimation when compared to diagnostics. Therefore, the optimal transition rule should reproduce the current land-cover state. In addition, multiple estimates of land-cover and carbon density from independent studies were employed to reduce the inherent uncertainties of diagnostics. Six widely used global land-cover datasets were integrated into an average map aiming to reduce the uncertainties that stem from a particular model or sensor observation. Similarly, for

assessing carbon stock, two different and independent datasets were collected.

It is important to note that the determined transition rules strongly depend on the land-use change dataset and the diagnostics used for evaluation. Rule 1 and rule 2 only serve to translate the changes for the LUH2 dataset to land-cover change. These two rules provide the best match of forest cover and terrestrial carbon stock from LUH2. This evaluation was only based on two critical properties of land-cover (i.e. forest cover and carbon stock) due to their significance in the exchange of water,



mass, and energy between atmosphere and land surface. In addition to carbon stock, the dynamics of forest cover from past to present highly interact with climate change, which is not considered in this study. In addition, the forest area and the LULCC carbon emissions and carbon density as resulting from LUH2 and the transition rules in this study are based on a simple global terrestrial model (i.e. Miami-LU model) and its uncertainties. Although the Miami-LU model includes the spatial heterogeneity

in vegetation regrowth rate and tracks subgrid-scale heterogeneity of carbon density in a manner similar to the more advanced Ecosystem Demography (ED) model (Hurtt et al., 1998;Moorcroft et al., 2001), the carbon emission estimates using the same transition rules and land-use change dataset would be different if other DGVMs or carbon accounting models were used. For example, the emissions from other studies in Table 3 may include emissions from soil pool decomposition, which is not accounted for in our model. In addition, rules other than 1 and 2 may produce better regional land-cover dynamics; new studies

aimed at determining continental-, country- or grid-specific transition rules are needed. Finally, the transition rules are defined as hard-clearing, meaning the vegetation would be totally removed or left totally intact. However, soft-clearing may be more realistic, in which part of the vegetation (quantified as the clearance ratio) is cleared. Future studies could focus on optimizing the clearance ratio using multiple land-cover type datasets.

This study determines an optimal rule that matched forest cover and carbon stock estimates from multiple vetted sources.

However, more research is needed to investigate the improvement of this rule on LULCC carbon emission estimates. To further reduce uncertainties in estimating land-cover dynamics, research could be expanded with emphasis on spatially and temporally varying rules. In addition to forest cover and carbon, more land-cover characteristics (e.g. forest age and tree height) are encouraged to be integrated to determine and constrain the optimal transition rules.

*Code and data availability*. The source code of GLM2 is available at http://luh.umd.edu/code.shtml, LUH2 dataset is available at http://luh.umd.edu/data.shtml. IPCC Tier biomass is available at https://cdiac.ess-dive.lbl.gov/epubs/ndp/global_carbon/carbon_documentation.html, Baccini aboveground biomass is available at https://daac.ornl.gov/cgi-bin/dsviewer.pl?ds_id=1337. TCCF, MODIS LC, GLCC, GFC, GLC2000 and GlobCover can be obtained from http://www.landcover.org/data/treecover/ , http://www.landcover.org/data/lc/ ,

https://edcftp.cr.usgs.gov/project/glcc/globdoc2_0.html , https://earthenginepartners.appspot.com/science-2013-global-forest/download_v1.6.html , https://forobs.jrc.ec.europa.eu/products/glc2000/data_access.php, http://due.esrin.esa.int/page_globcover.php respectively. All other data are available from authors upon reasonable request.

*Author contributions*. LM, GH, LC and RS designed this study. LM conducted the simulations and wrote the main body of the paper. All authors discussed the results and commented on the paper at all stages.

*Competing interests*. The authors declare that they have no conflict of interest.





*Acknowledgements.* We gratefully acknowledge the support to DOE-SCIDAC DESC0012972, and NASA-TE NNX13AK84A
and NASA-IDS 80NSSC17K0348.

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





**Figures & Tables**

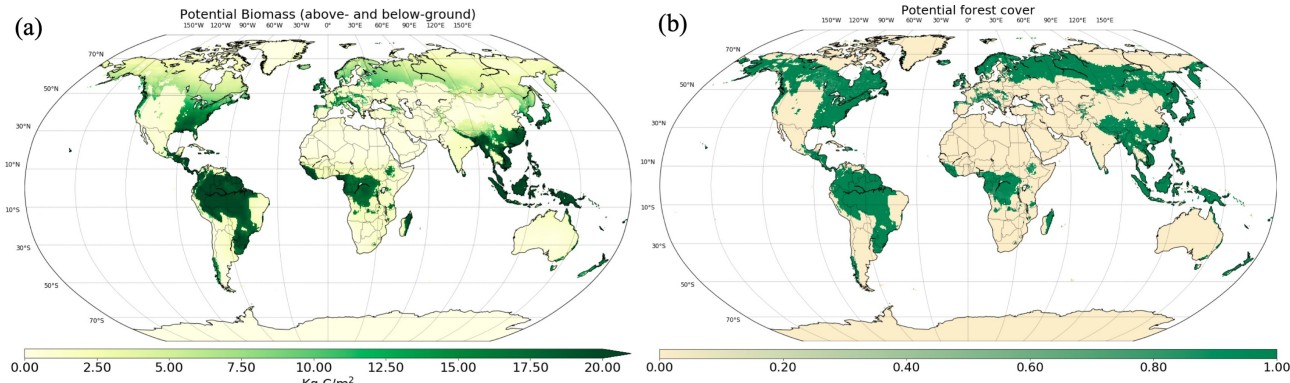

**Figure 1. Potential biomass density (a) and potential forest cover (b) in 850 estimated by Miami model.**





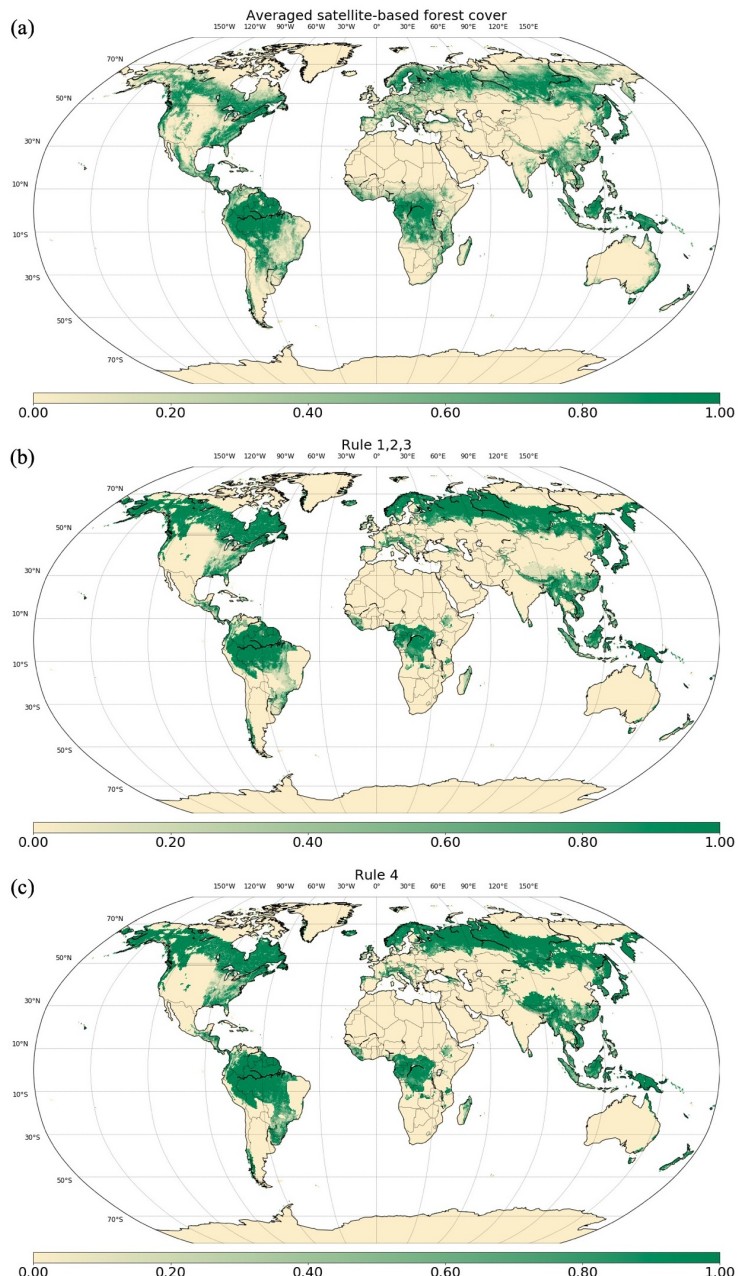

**Figure 2. Global forest cover in 2000 estimated by the 9 transition rules and the averaged satellite-based forest cover map. (a) Averaged satellite-based forest cover map; (b) Rule 1, 2, 3; (c) Rule 4.**





**Table 1. Rules for vegetation clearance during cropland, pasture and rangeland expansion. 'X' indicates complete removal of vegetation if the primary and secondary land state is altered. 'O' indicates no vegetation removal when land-use change occurs. 'F' indicates that vegetation is only removed if the preceding land cover is primary or secondary forest.**

| Transition Rule | Rule 1 | Rule 2 | Rule 3 | Rule 4 | Analytical rule 5 | Analytical rule 6 | Analytical rule 7 | Analytical rule 8 | Analytical rule 9 |
|---|---|---|---|---|---|---|---|---|---|
| ->Crop | X | X | X | X | X | X | O | O | O |
| ->Managed pasture | X | F | X | X | O | O | X | X | O |
| ->Rangeland | F | F | X | O | X | O | X | O | X |



**Table 2. Summary of land cover products used in this study including six satellite-based datasets and FAO FRA report.**

| Product | Global Forest Area (10⁶ km²) | Time | Publication | Data Type/Classification Scheme |
|---|---|---|---|---|
| GLCC | 40.89 | 1992-1993 | Loveland et al. 2000 | Land Cover (IGBP) |
| GLC2000 | 35.79 | 1999-2000 | Bartholome et al. 2005 | Land Cover (GLC 2000) |
| GlobCover | 37.38 | 2004-2006 | Bicheron et al. 2008 | Land Cover (GlobCover) |
| MODIS LC | 38.60 | 2001 | Friedl et al. 2010 | Land Cover (IGBP) |
| 1 Kilometer Tree Cover Continuous Fields (TCCF) | 42.74 | 1992-1993 | DeFries et al. 2000 | Tree Percentage |
| Global Forest Change (GFC) | 41.93 | 2000 | Hansen et al. 2010 | Tree Percentage |
| FAO | 40.55 | 2000 | FRA 2015 | National Censuses |





**Table 3. Summary of carbon emissions due to LULCC from available studies at pre-industrial and industrial period.**

| Reference | Time span | Carbon Emissions (Pg C) | LULCC types |
|---|---|---|---|
| *Pre-industrial Period* | | | |
| Reick et al., 2010 (bookkeeping model) | 1100-1850 | 80 | |
| Reick et al., 2010 (DGVM) | 1100-1850 | 47 | Cropland/Pasture Change |
| Pongratz et al., 2009 | 850-1850 | 53 | Cropland/Pasture Change |
| Stocker et al., 2011 | until 1850 | 69 | Cropland/Pasture Change, Urban |
| *Industrial Period* | | | |
| Houghton 2010 | 1850-2005 | 156 | Cropland/Pasture Change, shifting cultivation in tropics, and wood harvest |
| Houghton and Nassikas, 2017 | 1850-2015 | 145 | Cropland/Pasture Change, shifting cultivation in tropics, and wood harvest |
| Shevliakova et al.,2009 | 1850-2000 | 164 - 188 | Cropland/Pasture Change, shifting cultivation in tropics, and wood harvest |
| Pongratz et al.,2009 | 1850-2000 | 108 | Cropland/Pasture Change |
| Reick et al.,2010 (bookkeeping model) | 1850-1990 | 153 | Cropland/Pasture Change |
| Reick et al.,2010 (DGVM) | 1850-1990 | 110 | Cropland/Pasture Change |
| Stocker et al., 2011 | 1850-2004 | 164 | Cropland/Pasture Change, Urban |
| Le Quéré et., 2018 | 1850-2014 | 195 | Cropland/Pasture Change, shifting cultivation in tropics, and wood harvest |





**Figure 3. (a) Global forest area resulting from transition rules from 850 to 2015; (b) Comparison of global forest area in 2000 between remote sensing and FAO (shown as black bars) and results of transition rules (colored bars); (c) Annual change rate from 1850 to 2000. Positive value indicates the forest loss.**





**Table 4. Forest area ($10^6$ km$^2$) in 2000 of eight countries with the largest forest area, and all other countries combined ('Others'), estimated by the 9 transition rules, range compiled from satellite-based datasets and FAO report.**

| Country | Forest Area ($10^6$ km$^2$) | | | | | | | Range from satellite-based products and FAO |
|---|---|---|---|---|---|---|---|---|
| | Rule 1, 2, 3 | Rule 4 | Analytical rule 5 | Analytical rule 6 | Analytical rule 7 | Analytical rule 8 | Analytical rule 9 | |
| Russian Federation | 8.76 | 9.18 | 8.84 | 9.27 | 9.05 | 9.48 | 9.13 | 6.41-8.44 |
| Brazil | 4.63 | 5.70 | 4.90 | 5.98 | 5.07 | 6.14 | 5.34 | 4.21-5.95 |
| Canada | 5.62 | 5.67 | 5.63 | 5.67 | 5.80 | 5.84 | 5.80 | 3.41-4.41 |
| United States of America | 2.83 | 2.96 | 3.08 | 3.21 | 3.65 | 3.78 | 3.90 | 2.53-3.19 |
| China | 2.05 | 3.23 | 2.45 | 3.62 | 2.46 | 3.64 | 2.86 | 1.43-2.07 |
| Democratic Republic of the Congo | 1.57 | 1.61 | 1.60 | 1.64 | 1.63 | 1.67 | 1.66 | 1.57-2.12 |
| Indonesia | 1.32 | 1.34 | 1.37 | 1.40 | 1.60 | 1.62 | 1.65 | 0.99-1.70 |
| Peru | 0.76 | 0.78 | 0.78 | 0.80 | 0.77 | 0.79 | 0.79 | 0.70-0.80 |
| Others | 9.88 | 11.33 | 10.72 | 12.17 | 11.49 | 12.93 | 12.33 | 11.43-16.75 |
| **World** | **37.42** | **41.80** | **39.38** | **43.76** | **41.52** | **45.89** | **43.48** | **35.79-42.74** |



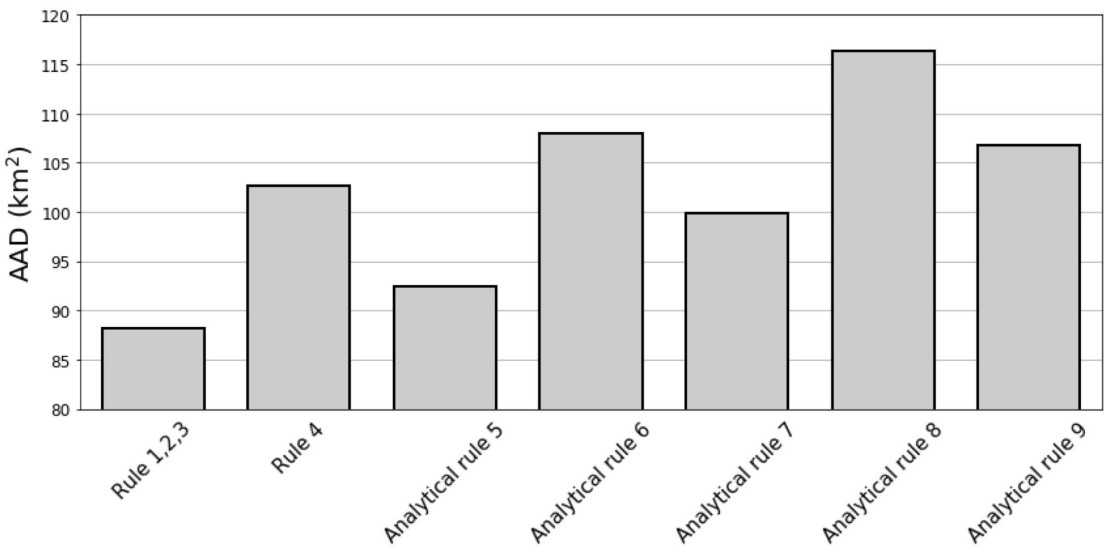

**Figure 4. Average of absolute difference in global forest area between maps estimated by transition rules and the averaged satellite-based forest cover map**.





**Table 5. Summary of LULCC carbon emissions estimated by the 9 transition rules and those from other studies in Table 4**

| Transition Rule | Carbon Emissions Estimation (Pg C) | | | Emission Range from Table 4 | | Estimation using LUH1 |
| --- | --- | --- | --- | --- | --- | --- |
| | 850-1850 | 1850-2000 | 1950-1960 | 850-1850 | 1850-2015 | 1950-1960 |
| Rule 1 | 72 | 175 | 20 | | | |
| Rule 2 | 70 | 170 | 19 | | | |
| Rule 3 | 77 | 185 | 22 | | | |
| Rule 4 | 55 | 142 | 16 | | | |
| Analytical rule 5 | 63 | 146 | 17 | 47-80 | 108-195 | 26 |
| Analytical rue 6 | 41 | 104 | 11 | | | |
| Analytical rule 7 | 28 | 107 | 13 | | | |
| Analytical rule 8 | 5 | 65 | 7 | | | |
| Analytical rule 9 | 13 | 67 | 7 | | | |



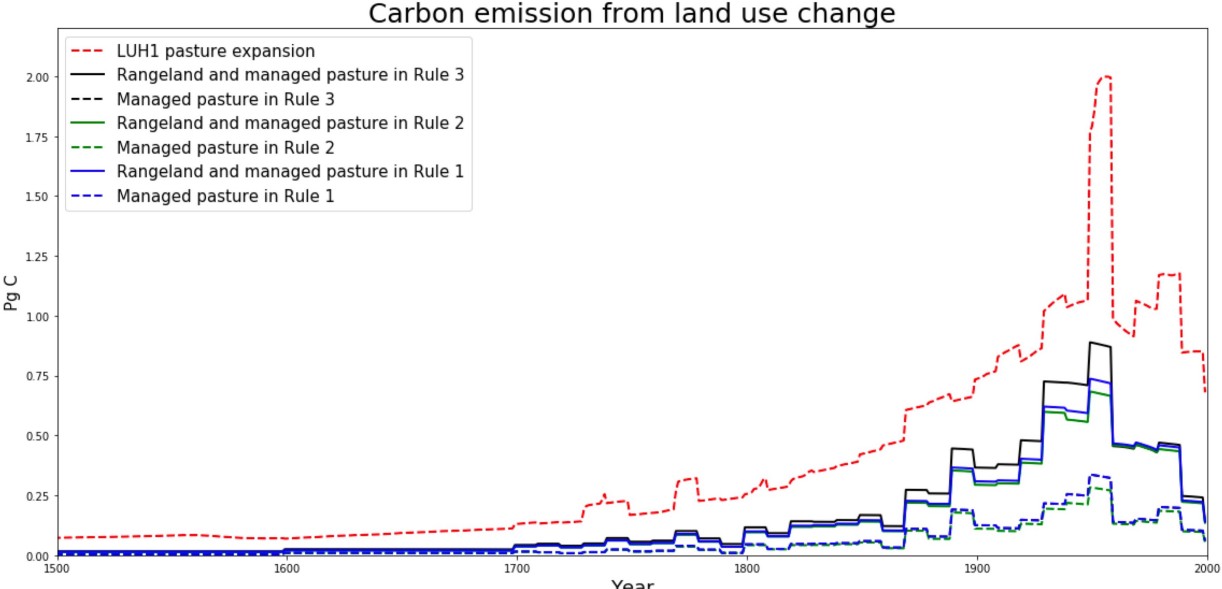

**Figure 5. Carbon emission due to vegetation (forests and non-forests) removal in expansion of managed pasture and rangeland. Red dash line represents emission from pasture expansion in LUH1. Blue, green and black solid lines represent emission from expansion of managed pasture and rangeland in LUH2 estimated by rule1, 2 and 3 respectively, and blue, green and black dash lines are emission from managed pasture expansion only by rule1, 2 and 3 respectively. Note that the pasture category in LUH1 corresponds to managed pasture and rangeland together in LUH2.**

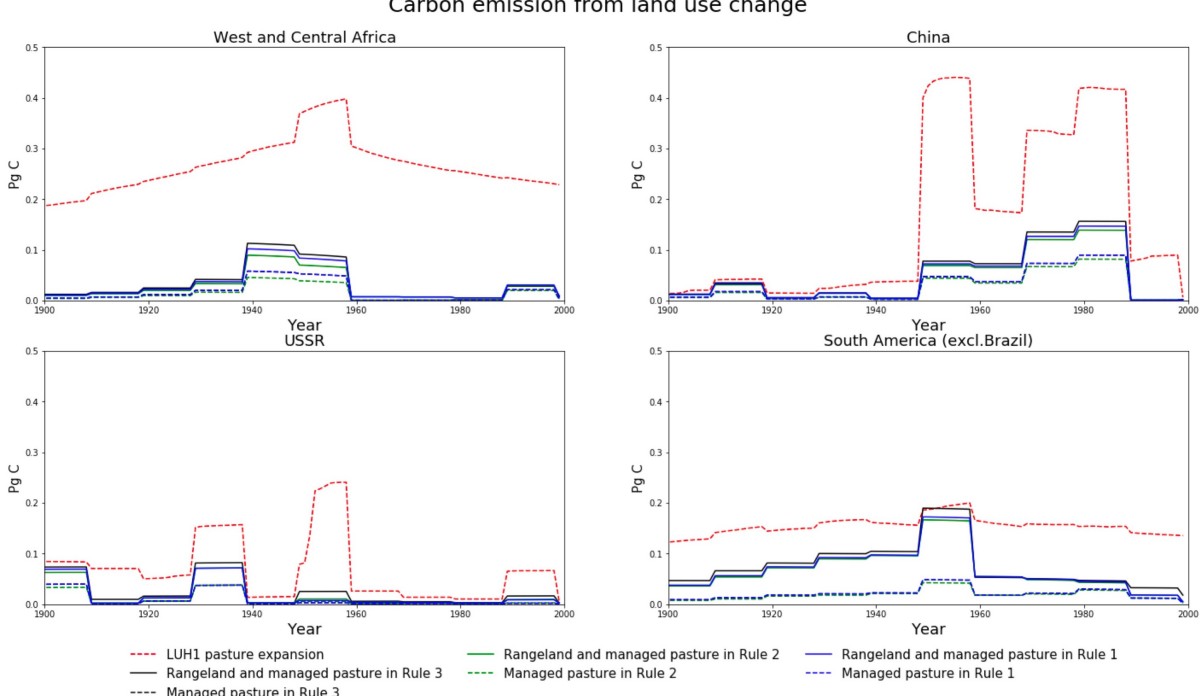

**Figure 6. As in Figure 5 but four regions and countries: (a) West and central Africa; (b) China; (c) Russian Federation; (d) South America.**





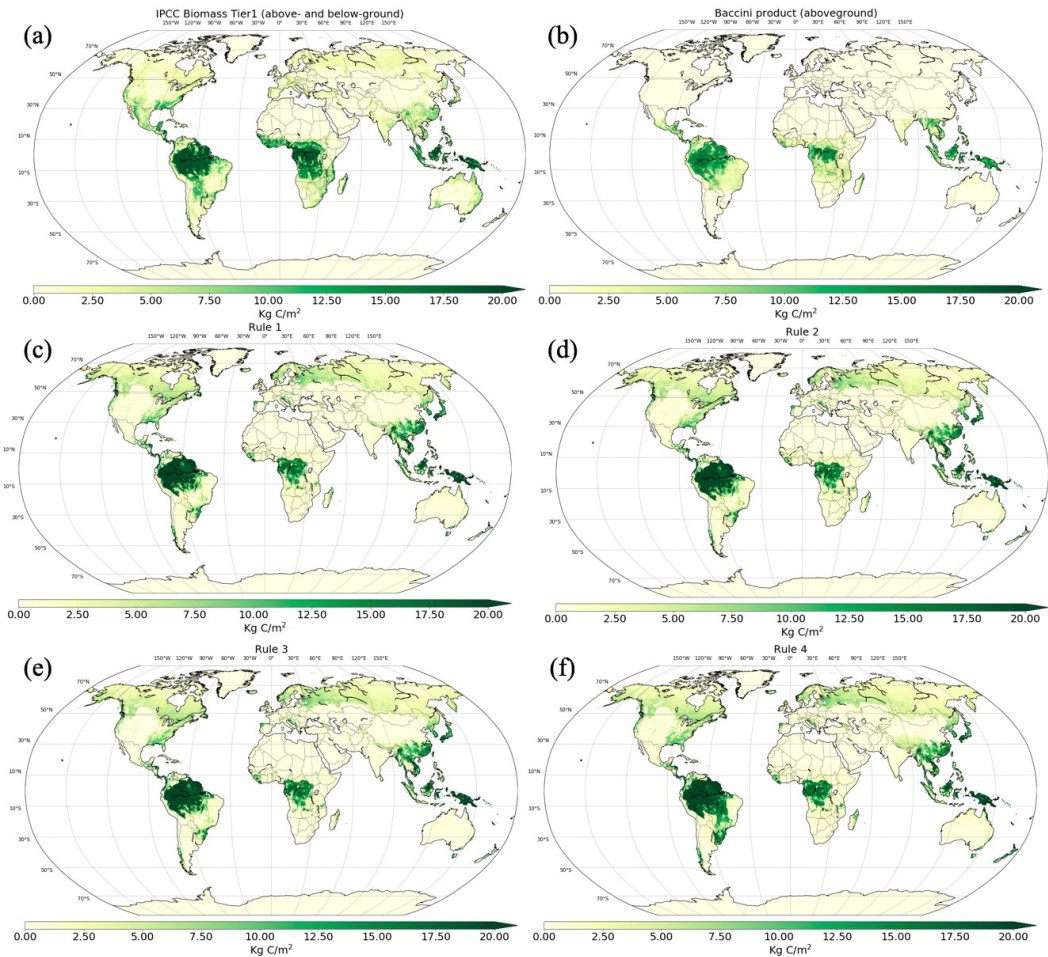

**Figure 7. (a) IPCC Biomass Tier 1 density; (b) Baccini's product (only aboveground) at pantropical; global carbon density (above- and below-ground) maps estimated by rule 1-4 from (c) to (f).**





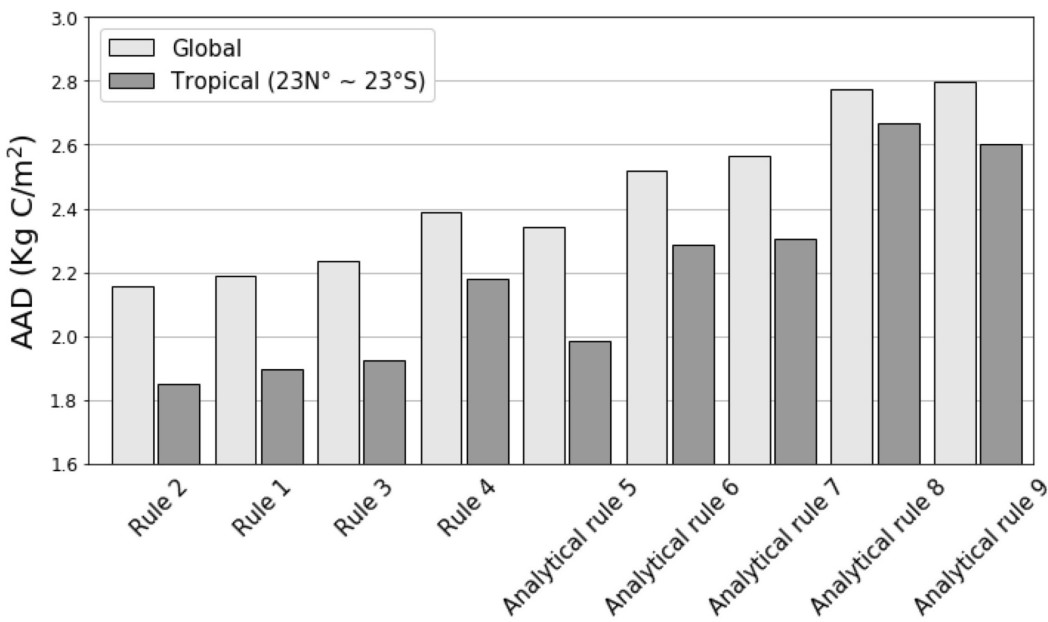

**Figure 8.** Average of absolute difference in carbon density between estimations of the 9 transition rules and two diagnostic maps: global comparison with IPCC biomass density map (incl. above- and below-ground); tropical comparison with Baccini's carbon density map (only aboveground).



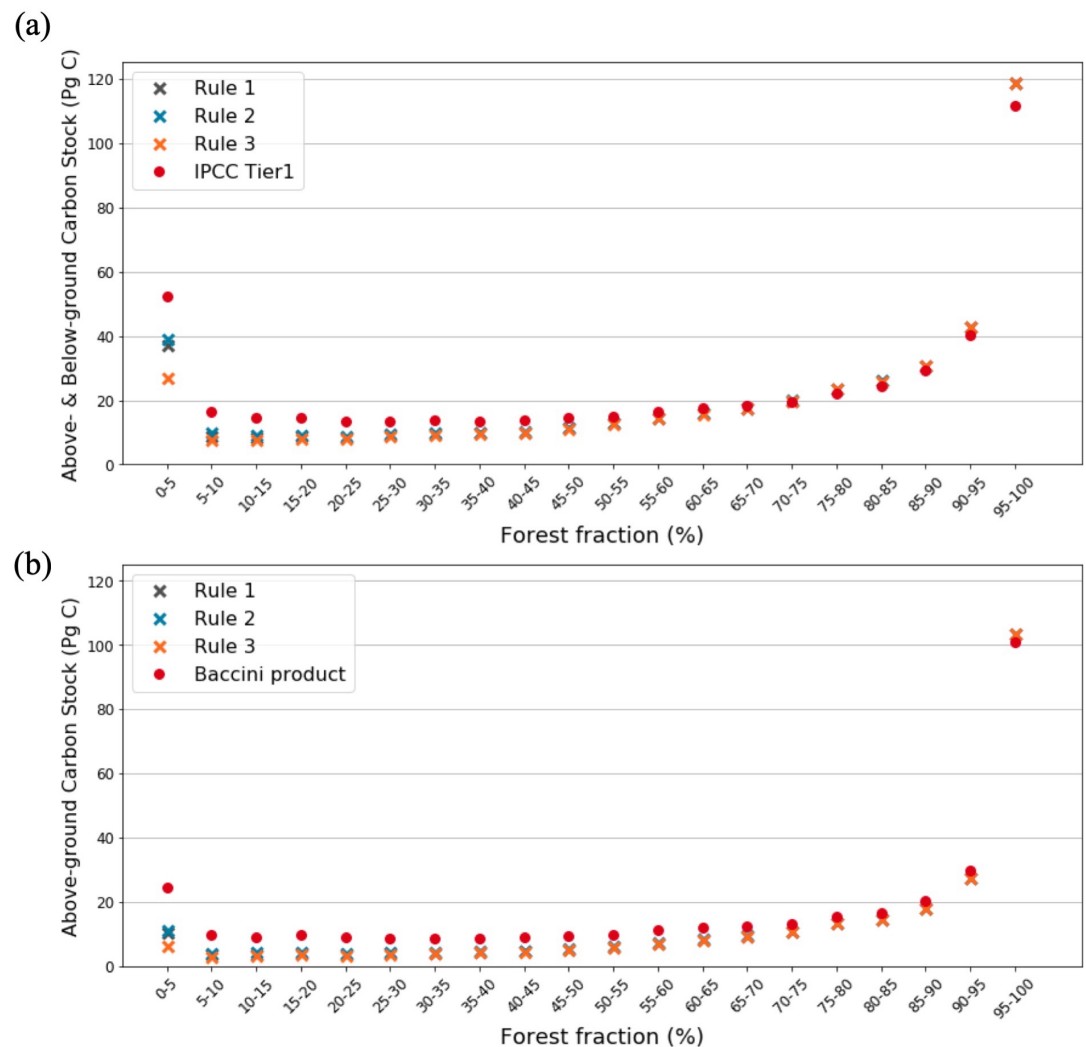

**Figure 9. Total carbon stock grouped by forest fraction from averaged satellite-based forest cover map. (a) global; (b) pantropical.**