# Peer review of "Global Rules for Translating Land-use Change (LUH2) To Landcover Change for CMIP6 using GLM2"

_Geoscientific Model Development, 2019_

## Referee Comment (RC1) · Anonymous Referee #1 · 21 Aug 2019

Review of gmd-2019-146: Global transition rules for translating land-use change (LUH2) to land-cover change for CMIP6 using GLM2

Summary

The authors present an analysis of how various land conversion rules affect forest area and biomass carbon over the historical period, when applied to the LUH2 land use data. The results are compared with several available data sets, and they find that while a few different rules may be reasonable, the rule corresponding to definitions of source land use data appears to be the most reasonable one to use. They conclude by recommending full clearing of vegetation for cropland, urban, and managed pasture

and clearing only for rangeland, when applying LUH2 data to ESMs and DGVMs.

Overall response

I, and I think many others, will be happy to see these results published. This is a necessary step toward improving estimates of LULCC effects on the earth system. The paper is relatively clear, but some additional clarification and discussion are needed. Please see the details following these main concerns:

1) The methods describing how vegetation fraction and land-use fraction are tracked (section 2.3) are not complete.

2) While there is some discussion of "forest area" vs "forest/tree cover," it is still unclear which metric is being used and discussed in the comparisons.

3) Please include discussion of the analytical rule results. You present the numbers, but do not tell us what they mean in terms of how particular transitions affect forest area and carbon dynamics.

4) It would be useful to see regional comparisons for carbon.

Specific suggestions and comments:

Abstract

page 1, line 23: "optimal" is a strong word here for a global transition rule. "most reasonable global transition rule. . ." may be more appropriate

Introduction

page 2, lines 9-10: this is a misleading statement not supported by the inappropriate reference (which is also quite old). it isn't necessarily the case that recovering or planted or re-planted forest has lower potential biomass many plantations or managed forests have higher growth rates than unmanaged forest, and in time could easily match or outgrow unmanaged biomass levels furthermore, the reference is about avail-

Interactive
comment

able land for afforestation, and does not compare afforested stands with corresponding primary stands. there are several recent papers that estimate carbon sequestration potential of forests. here is one example: griscom et al 2017: natural climate solutions in PNAS, oct 31 2017 vol 114 no 44 11645-11650

page 2, lines 18-22: awkward sentence that is difficult to read. split it up and bring the examples out of the parentheses

Methodology

page 5, lines 22-23: are there other factors not taken into consideration? is the climatology constant or does it vary with time? be specific here - state exactly what factors are or are not included

page 5, lines 26-29: what tree density or how much tree canopy cover will give 2 KgC/mˆ2? what defines potential forest area in the two comparison studies?

page 6, lines 5-28:

These equations are incomplete, and therefore confusing. Transitions from types 5-8 to 5-8 are not included for f-gained. As such we do not see that these types of gains account for the f/l ratios of the losing types 5-8. Transitions from types 1-4 to 1-4 are not included for losses or gains. Doesn't harvest move primary to secondary land? Are there any gamma factors for these transitions? Also, wouldn't abandoned ag land move to secondary first, then from secondary to primary? For this path from ag to secondary to primary, how is the abandoned ag vegetation fraction tracked over time?

Furthermore, there are corresponding equations for l(i,t+1), correct? But they are different because they track the land use transitions upon which the vegetation changes (without the gamma parameter that alters vegetation cover). This distinction and relationship between the two needs to be made clear.

page 6, lines 20-21: f(i,t) could also be equal to l(i,t)

page 6, lines 22-23: "these" is confusing. maybe delete "these" and make "land-use" at the end plural.

page 7, line 21: considering annual creation and discard of wood products and decay in landfills, these emissions can grow to be quite large over time.

page 7, lines 22-32: This does not appear to account for different areas of secondary land within a grid cell with different ages. Similarly to my comment above, if cropland is abandoned and has half natural secondary veg and half crop veg, there can be at least three different ages of secondary land in the cell. The number of different ages will grow with each transition. How is this dealt with? When does secondary become primary land again? Does primary land always have biomass density of B0?

page 8, line 25: do you expect a 30% tree cover threshold to correspond with GLM's 2 KgC.mˆ2 threshold? Also, are the >30% pixel areas used for the values in table 2, or are the values in table 2 pre-threshold tree cover fractions that were calculated?

page 10, lines 4-28: What numbers are you comparing here and in figures 3-4 and table 4? Are the six-dataset numbers the tree cover or the forest with cover >30%? What does GLM's biomass threshold represent in relation to these metrics? Also, there are a couple of other tree cover estimates that are much lower: see meiyappan and jain 2012 frontiers in earth science 6(2):122 and Li et al 2018 earth system science data, 10:219.

page 11: Can you make regional comparisons of LULCC emissions with info from the sources in table 3?

page 11, lines16-25: It would be helpful to see regional comparisons of carbon stock here.

page 12, lines 11-13: up to this point you report that rules 1-3 have the same forest area dynamics, which makes sense because forest is always cleared in all three rules. but here you state that there are differences between rules 1 and 2 for forest area.

[Figure]

please clarify why this is the case, and if they are different it needs to be noted early on, even if the differences are negligible.

page 12, lines18-30: This is repetitive of your methods. It would be more useful to have discussion of the alternative rules, which you include in your figures/tables, but do not comment on at all.

Figures and Tables

Figure 2 Only the results for rules 1-4 are shown, not all 9 as in the caption

Table 5 Do you mean compared with studies from table 3?

Figures 5 and 6 I don't see the black dashed line. is it hidden?

Figure 8 I suggest plotting them in rule # order - rules 1 and 2 appear to be switched

---

## Referee Comment (RC2) · Eddy Robertson (Referee) · 23 Aug 2019

The paper provides a useful assessment of how the choice of implementation of land-use data can affect the simulation of tree cover and carbon emissions. A more detailed description of the model used is needed to allow better interpretation of the results. A clearer justification of the choice of optimal transition rule is needed. Key uncertainties in the model simulation need to be discussed, in order to assess whether the results show the optimal transition rule for all models or simply for GLM2. I have also suggested a few minor corrections.

**Major Corrections:**

- P5, L21: Please provide more detail about the Miami model and the simulation of carbon stocks.

  – What inputs does the Miami model use? Does it use climate data? What is the MSTMIP climatology?

  – Is the Miami model a process-based model or a statistical model?

  – What time period is being simulated? What period does the MSTMIP climatology cover? Does the model use $CO_2$ concentration from the year 850, or is $CO_2$ concentration simply not a factor the model can consider?

- Discussion: The paper should comment on the significance of uncertainty in the map of potential carbon stocks, for example if the global total potential carbon stock were only 557 Pg C, do you think a different transition rule would be optimal?

  – Related to the above point, is whether different rules are best for different ESM, because they will simulate different potential carbon stocks. Please comment.

- P5, L26 and discussion: Why was 2 kg C/m2 chosen to define a forest? Is using 2.2 kg C/m2 equally justifiable and if so, might this lead to a different transitional rule being optimal?

- P10, L27 and figure 4: Disagreement with the average satellite-based forest cover does not mean that the model is not consistent with (i.e. within the range of) the ensemble of satellite-based forest cover. Please can you at least mention this possibility. You could account for this uncertainty, perhaps by adding an uncertainty bar to figure 4 showing the errors relative to the TCCF and GLC2000 datasets.

- Figures 5, 6 and 9: Please add results from rule 4 to figures 5, 6 and 9. I don't know why you have stopped considering rule 4.

- Discussion: More justification is needed for you choice of optimal rule. I agree that, all else being equal, using rule 1 is best because it is consistent with HYDE 3.2. It's fair to exclude rule 4 because it produces too much tree cover (table 4) and compared to the other rules it has twice the carbon stock error in the tropics (figure 8). However, in the discussion you need to more clearly state why rule 4 is excluded. I don't know why rule 3 has been excluded, please can you justify this choice?

- Discussion: Will the choice of rule matter less in future simulations? Is there less range-land expansion in future?

**Minor Corrections:**

P2, L18-22: This sentence needs to be made clearer. I also think that you are underselling the importance of LUH2, it is not only used in land-use specific model simulations, it is a key input to the DECK and historical simulations as well as for future projections (scenarioMIP). Additionally, you could mention that it is used to simulate the biophysical effects of land-use change as well as the biogeochemical effects.

P3, L11-15: This sentence needs to be made clearer.

P8, L1: What is the "wood fraction?"

P13, L4: Please remove reference to the "Miami-LU model" and replace with "Miami model" or "GLM2," as appropriate.

Caption of Table 5: Should refer to table 3 not table 4?

---

## Referee Comment (RC3) · Anonymous Referee #3 · 16 Sep 2019

**General comments**

The manuscript entitled *Global Transition Rules for Translating Land-use Change (LUH2) To Land-cover Change for CMIP6 using GLM2* by Ma *et al.* aims at recommending a global transition rule for translating LUH2 land-use forcing into land-cover changes in CMIP6 models. The authors simulate land-use induced land-cover changes based on a set of translation rules using the GLM2 model and prescribed LUH2 land-use transitions. Subsequently, emerging present-day forest cover, biomass density, and LUC emissions are evaluated against published estimates. The authors conclude by recommending a rule where all vegetation is cleared upon cropland/pasture expansion and only forest vegetation is cleared upon rangeland expansion.

The paper is technically correct, well written, timely in providing recommendations how to translate land-use forcing into land-cover changes for CMIP6 and the content is generally suitable for publication in GMD. I am wondering, however, if the conceptual design of the study is valid in the context of the large uncertainties that exist in the reconstruction of historical land use. Likewise, I do not think that the main conclusion ('optimal' rule being rule 1) is supported by the results presented. The wording ('optimal transition rule', 'accurate quantification', […]) is not suitable in the context of large uncertainties. Rather than claiming an 'optimal' transition rule, I suggest a framing towards recommending a 'consistent' translation rule for implementation of land use in ESMs/DGVMs. A consistent treatment in ESMs/DGVMs would eliminate added uncertainty and complexity from different treatment in each model. If this is 'optimal' (or and 'optimal' global rule even exists), is difficult to judge with available data and not shown by the contents of this manuscript. The reproducibility of the analysis presented is questionable, especially as neither the code of GLM2 (though stated in the *code and data availability* section) nor any documentation of the model is publicly available at the moment.

**Major comments**

**Conceptual design.** One of the main (implicit) assumptions in the manuscript is that the land-use transitions from LUH2 are 'correct', although these transitions are affected by large uncertainties and necessarily based on many assumptions. Given also the presented results that show simulated forest cover (and mostly also carbon emissions) in the range of previously published results for rules 1-4 (and partly even for the 'analytical' rules), I am wondering how valid any conclusions drawn regarding an 'optimal' translation rule can be. Without evaluation if any of the reference estimates is 'better' than others, I cannot see why Rule 1 is more 'optimal' than Rule 2, 3, or 4 (as long as they are all within the range). Moreover, if we can see only this small discrepancy already with assuming LUH2 data as 'correct', how do the authors think, would this evolve, if accounted for uncertainties in the land-use reconstruction? For example, how would the LUH2 high and low estimates change the results? How would prioritizing another land-use type in the allocation of land-use transitions in GLM2 (if this still exists like in GLM) change the results? And in conclusion: Does it even matter which of the rules is applied in an ESM/DGVM given these probably much larger effects from the mentioned uncertainties (besides of being consistent across CMIP6 models)?

**Wording/Framing.** Closely related to the comment above, I do not think that the wording in the manuscript is appropriate at many instances. The authors should avoid terms like 'accurate', 'optimal', etc. in the context of historical land-use change and its translation to land-cover change. As the authors state correctly in their discussions, globally valid transition rules probably do not exist and I would encourage the authors to rather emphasize the underlying uncertainties than trying to hide them behind strong words. Instead of claiming to derive 'optimal' rule(s), it would be more useful to recommend a 'reasonable rule' that should be used consistently across CMIP6 models.

**Conclusions.** The main conclusion (=recommending rule 1) is not supported by the results presented. For all of the proposed diagnostics (forest cover, biomass density, carbon emissions), all of the rules (1-4) are within the range of the diagnostics (sometimes even some of the analytical rules which are supposed to be idealized/unrealistic), and all of them are far from 'accurately' translating land-use change to land-cover change as the authors claim. The fact that on average one of the rules is closer to an averaged reference map does not provide justification that one of the four rules is superior over the others. The authors need to provide more justification why they recommend rule 1 and/or 2 based on the results presented here. Personally, I do not think this is possible without either defining a 'best/most suitable' reference map (e.g., based on GLM2 forest definition) or extended (spatial) analysis to identify regional characteristics of the different rules. Generally, the

discussion/conclusion section needs to be strengthened, as in its current form it mainly repeats methods/results instead of discussing the findings of the analysis.

**Specific comments**

**General.**
Please check the whole manuscript for missing whitespaces in front of references and within references.

**Abstract.**

P1 L15 'accurately' does not seem to be an appropriate wording given the large uncertainties both in climate and land-use modeling. Please remove.

P1 L16 I would suggest to use 'land-cover change time series' only, i.e. remove the *'land-cover'*.

P1 L18-21 Please include that GLM2 was used for the simulations already here.

P1 L23-25 I think 'optimal transition rule' is not the correct wording.
This sentence is also quite complicated. I would suggest to rephrase, emphasizing that within GLM2 the mentioned rule turned out to perform best. The wording here indicates that this rule is 'optimal' irrespective of the model used, which is not supported (and probably also not intended) by the results of the manuscript.

P1 L26-28 I am wondering if mentioning the detailed forest area is required, while not referring to the carbon density and LUC emissions at all?

**Introduction.**

P2 L5-7 I think it is not correct to present this statement as a fact. The numbers are based on a historical LU reconstruction, i.e. model results. Please rephrase to, e.g., *'Model results show […]'* or *'It has been estimated, […]'*.

P2 L7 Include *'amongst others'* as there are many more impacts of LUC and land management on the carbon cycle than deforestation, afforestation, and wood harvest.

P2 L12-13 Same as above. Please highlight that this is also an (uncertain) model result, e.g. by including the uncertainty ranges.

P2 L13-14 What are *'these emissions'*? Please rephrase in a way that the reference to land-use emissions becomes clear.

P2 L13-15 The numbers presented here are probably all derived by using LUH(2) as land-use forcing. Thus, I think it would perfectly fit the storyline to add a sentence that explains that exactly due to these uncertainties a 'better' translation between land use and land cover is required. Otherwise, one may ask, why we would need all these transition rules, if we already know about historical land-use impact.

P2 L18-22 What about just saying
*LULCC reconstructions enter Earth System Models (ESMs) (e.g., Lawrence et al., 2016), Dynamic Global Vegetation Models (DGVMs) (e.g., Le Quéré et al., 2018), and bookkeeping models (Hansis et al., 2015; Houghton and Nassikas, 2017) to quantify biogeochemical and biophysical impacts of historical land-use change.'*
I don't think the details about models and MIPs is required here.

P2 L18-22 Remove *'(e.g., HYDE, SAGE)'*. Replace *'Goldewijk et al., 2017'* by *'Klein Goldewijk et al., 2017'*. Add *Ramankutty and Foley, 1999* and *Pongratz et al., 2008.*

Ramankutty, N. and Foley, J. A.: Estimating historical changes in global land cover: Croplands from 1700 to 1992, Global Biogeochem. Cycles, 13(4), 997–1027, doi:10.1029/1999GB900046, 1999.

Pongratz, J., Reick, C., Raddatz, T. and Claussen, M.: A reconstruction of global agricultural areas and land cover for the last millennium, Global Biogeochem. Cycles, 22(3), GB3018, doi:10.1029/2007GB003153, 2008.

| | |
|---|---|
| P2 L24-26 | The manuscript is about historical land use, i.e. the harmonization with future LULCC seems to be an irrelevant information here. |
| P2 L28-33 | Remove the reference '*Shevliakova et al., 2013*' between the sentences and only put it in the end of the paragraph. |
| P3 L1-3 | I would not agree that there is '*a lack of explicit global rules*'. As the authors show later on, it is relatively easy to come up with some. I would rather argue that there is no consistency/agreement on which rule to apply. Apart from that, in this context it is also worth to mention that such 'global transition rules' probably do not exist at all [see, e.g., Prestele *et al.* 2017]. |

Prestele, R., Arneth, A., Bondeau, A., de Noblet-Ducoudré, N., Pugh, T. A. M., Sitch, S., Stehfest, E. and Verburg, P. H.: Current challenges of implementing anthropogenic land-use and land-cover change in models contributing to climate change assessments, Earth Syst. Dyn., 8(2), 369–386, doi:10.5194/esd-8-369-2017, 2017.

| | |
|---|---|
| P3 L5 | … '*and the location where a land-use change happens*'. |
| P3 L5-7 | While this statement sounds very intuitive, I wonder if there is any literature supporting these tendencies? |
| P3 L11-15 | Complicated sentence. Please shorten.
In my opinion, everything after '*…not yet provided*' is not necessarily required. What about joining with the following sentence instead? Isn't it exactly what the authors are aiming at: providing recommendations how to treat these 'new land-use types' in the translation?

Suggestion:
'*However, explicit suggestions for land-cover and carbon stock modifications resulting from these new defined land-use types are not yet provided, but are crucial for the translation of land-use change to land-cover change within ESMs or DGVMs. An inconsistent translation will potentially produce very different land-cover dynamics, which will impact the land surface biophysical and biochemical processes.*' |
| P3 L18 | I would not agree that the approach presented here will reduce any uncertainty. It rather can provide recommendations for consistent treatment across models, if the 'optimal' rule is adapted by the CMIP6 models. But this does not allow any conclusions how uncertainty will be affected. |
| P3 L22 | Remove '*other*' in front of '*independent*'. |
| P3 L22-25 | Here, too, I recommend not using the term '*optimal*'. |

**Methodology.**

| | |
|---|---|
| P4 L1-5 | Is there a specific reason not to include the ESA CCI land cover for comparison?

Remove the details about the comparison dataset here. They are all mentioned in section 2.5. |
| P4 L7-28 | As the method section is already quite long, I would suggest to shorten here. I do not think there is a lot of added value to describe LUH2 in this detail for the purpose of the paper. I guess |

there will be an associated LUH2 publication soon, so it is probably enough here to just describe the key features that are relevant for the analysis in this manuscript.

P4 L11-13    This sentence doesn't seem to fit in the context here.

P4 L17       While '*data-driven*' is probably not wrong, I think it is misleading as it implies that the constraints used in LUH2 are based on observations. However, to my knowledge most of the constraints are model outputs in some way (be it the HYDE reconstruction or models derived from remote sensing images, etc.). Therefore, I would recommend not to use '*data-driven*' here.

P4 L28       Where do the 2 kg C/m² come from? How do they relate to other forest definitions? Are there any references that could support this threshold? Some more information would be valuable for the reader.

P5 L1        What are the '*analytical purposes*' of rules 5-9? For the rest of the manuscript they are mostly used to state that the results with these results are 'way off', but this is not very surprising given their idealized/unrealistic character. I would therefore recommend to leave them out, as they rather add confusion.

P5 L8-12     I agree that the effect of spatial and temporal varying rules is beyond the scope of this study. However, these are very strong simplifications and it would be useful to get an indication of how including this variation would affect the results. Maybe the authors could look a bit more detailed into the country-level and gridded results for the different rules and diagnostics. Can there be seen any patterns, if one of the rules is 'more likely' in certain regions than in others? If this is not feasible, the authors should include more detailed elaboration how the results may be affected in the discussion section.

P5 L18-19    It sounds a bit 'circular' that the output of GLM2 (i.e., LUH2) is used as input into GLM2 for the analysis in this manuscript. Could the authors provide more explanation how this was implemented? Are these independent model runs?

P5 L23-24    More detail required regarding the model run (time period, etc.).

P5 L23-29    These are effectively 'results'. I would recommend to move to section 3.1.

P5 L31 ff.   I do not know about the specifications of GLM2/LUH2, but in LUH1 [Hurtt *et al.* 2011], choices had to be made about starting date, priority for land-use transitions, wood harvest inclusion, etc. If this still exists for GLM2/LUH2, it would be useful to indicate here, which configuration of GLM2 was used to derive LUH2 to allow the reader to understand how the historical transition rates have been derived. In the discussion, a short evaluation of how changing these assumptions would change the results of the analysis, would help.

Hurtt, G. C., Chini, L. P., Frolking, S., Betts, R. A., Feddema, J., Fischer, G., Fisk, J. P., Hibbard, K., Houghton, R. A., Janetos, A., Jones, C. D., Kindermann, G., Kinoshita, T., Klein Goldewijk, K., Riahi, K., Shevliakova, E., Smith, S., Stehfest, E., Thomson, A., Thornton, P., van Vuuren, D. P. and Wang, Y. P.: Harmonization of land-use scenarios for the period 1500-2100: 600 years of global gridded annual land-use transitions, wood harvest, and resulting secondary lands, Clim. Change, 109(1), 117–161, doi:10.1007/s10584-011-0153-2, 2011.

P7 L26       Not an expert in carbon impacts of land management. However, I am wondering if management of land necessarily means that there is no further accumulation of biomass in the remaining 'natural' vegetation?

P9 L20       Instead of only using an average 'smallest difference' for the gridded results, looking more into the spatial patterns would maybe help to derive stronger justification for recommending one of the suggested rules.

**Results.**

P9 L29    '*Higher forest cover*' compared to what? The average reference map? Unclear.
          In addition, rather than presenting the three forest cover maps in Fig. 2, it would be more useful to show difference maps (rule 1-3 minus reference; rule 4 minus reference). This would facilitate the identification of differences between the maps.

P10 L9-11  Here, the authors emphasize the uncertainty in the definition of 'forest', which cannot easily be resolved. Which definition used in one of the reference maps is closest to the forest definition used in GLM2 (> 2 kg C/m²). Using the 'closest' map to compare the GLM2 results to would probably give a better indication than having a huge range of 'reference' maps (where the range partly originates 'only' in definition issues).

P10 L12-15  Rule 7 is within the range according to Fig. 3.

P10 L16-17  Rule 5 and 7, too (see Fig. 3).

P10 L19-20  The statement is not wrong, but also the analytical rules 'locate' around 75% of global forest land in these eight countries. This cannot be used as a characteristic of distinction between the rules.

P10 L27-28  I am not sure, if the mean average delta compared to an average global forest map is a good metric here. The average reference map is a rather 'artificial' map, and not necessarily the most plausible one. Does the assessment of 'smallest' difference change, if compared to the reference maps individually?
          Rather than using averages, I think the authors should aim at identifying a reference map that corresponds most with the forest definition within GLM2 and compare to this map.
          Additionally, showing a map of the differences would also allow to identify if certain rules match the current situation better in particular regions. As several rules are within the range of published forest cover areas, this would allow a better justification for one certain rule and/or regional diversification.

P10 L10-12  What happened to rule 4? From Table 5 it can be seen that it reduces the pasture anomaly, too. Why does it not appear any more in the text and Figs. 5/6?

P10 L13    Is the difference of 1 Pg C really a 'significant' difference?

P11 L16-20  The differences in the average difference between model and reference are rather small across all rules (except for some of the analytical). Again, are the authors sure that this average difference is a suitable indicator (see also comment above reg. forest area)?

P11 L22-25  In Fig. 9 hardly any difference can be seen for the three rules. How large are the differences between the individual rules and the reference? How do the authors conclude from this Fig. that rule 1 and 2 are closer than rule 3? What happened to rule 4?

**Discussion and Conclusions.**

P11 L28-30  Please see major comment 'Conclusions.'

P12 L1-5   These statements are not wrong, but only repeat parts of the results. Please remove.

P12 L7-8   How do the authors think that the results presented here can facilitate the reconstruction of historical land-cover change? Please elaborate.

P12 L10 ff.  It is still not clear at this point why rule 1/2 are 'better' than rule 3/4. Additionally, I wonder what is the added value of the study, if one of the main conclusions is that rule 1 is 'better' than rule 2 due to assumptions taken in HYDE.

| P12 L24-30 | While not wrong, I think irrelevant here as it (1) mainly repeats what has been written in the methods section and (2) does not provide justification for one of the rules presented. The authors should aim at emphasizing the reasons why they recommend rule 1 to CMIP6 models. |
|---|---|
| P13 L6 | Not clear why the authors introduce a new model here. |
| P13 L14 | I do not agree with this statement/conclusion. Several rules presented here lead to similar results (within the range of reference maps) and justification is missing, why one of the rule is better than another. The claim that an 'optimal' rule has been determined by the analysis is not supported by the results. |

**Figures and Tables.**

| Figure 1 | Isn't Fig. 1(b) a binary map (forest/no-forest)? In this case the legend doesn't make sense. |
|---|---|
| Figure 2 | Please add difference maps to facilitate the identification of differences between the maps. Only 4 rules (instead of 9 as mentioned in the caption) are shown. |
| Table 1 | Are the 'analytical rules' required for the purpose of the manuscript? |
| Figure 3 | For the analytical rules the x-axis labels are not centered any more. |
| Figures 5-6 | Why is rule 4 omitted from these Figs.? |
| Figure 7 | Also here difference maps would help to guide the reader. |
| Figure 8 | Switch Rule 1 – Rule 2 (x-axis). |
| Figure 9 | Differences between the rules hardly can be seen. Maybe zooming in for different percentages would improve the readability? |

---

## Author Comment (AC1) · 5 Dec 2019

Responses to reviews for manuscript **Global Transition Rules for Translating Land-use Change (LUH2) To Land-cover Change for CMIP6 using GLM2**

**To Reviewer #1:**

**Reviewer 1:** The authors present an analysis of how various land conversion rules affect forest area and biomass carbon over the historical period, when applied to the LUH2 land use data. The results are compared with several available data sets, and they find that while a few different rules may be reasonable, the rule corresponding to definitions of source land use data appears to be the most reasonable one to use. They conclude by recommending full clearing of vegetation for cropland, urban, and managed pasture and clearing only for rangeland, when applying LUH2 data to ESMs and DGVMs.
Overall response I, and I think many others, will be happy to see these results published. This is a necessary step toward improving estimates of LULCC effects on the earth system. The paper is relatively clear, but some additional clarification and discussion are needed. Please see the details following these main concerns:

**Response:** Thank you very much for your comments which are very conductive to improve the manuscript. We have carefully revised the manuscript accordingly. Note that 'transition rule' has been renamed as 'translation rule' throughout the manuscript to avoid confusion with LUH2 land-use transitions. Please see our point-by-point response below.

**Reviewer 1:** 1) The methods describing how vegetation fraction and land-use fraction are tracked (section 2.3) are not complete.

**Response:** Please see details in relevant comment below.

**Reviewer 1:** 2) While there is some discussion of "forest area" vs "forest/tree cover," it is still unclear which metric is being used and discussed in the comparisons.

**Response:** Only forest cover/area were simulated and compared between model and reference datasets. We have clarified what metrics are used at the last paragraph of section 2.5.

**Reviewer 1:** 3) Please include discussion of the analytical rule results. You present the numbers, but do not tell us what they mean in terms of how particular transitions affect forest area and carbon dynamics.

**Response:** We have added discussions of Rules 5-9 at results section.

**Reviewer 1:** 4) It would be useful to see regional comparisons for carbon.

**Response:** Regional comparisons of carbon density estimate has been added as Figure S4.

**Reviewer 1:** Specific suggestions and comments: Abstract page 1, line 23: "optimal" is a strong word here for a global transition rule. "most reasonable global transition rule. . ." may be more appropriate

**Response:** We have changed 'optimal' to 'recommended'.

**Reviewer 1:** page 1, line 23: "optimal" is a strong word here for a global transition rule. "most reasonable global transition rule. . ." may be more appropriate

**Response:** We have changed 'optimal' to 'recommended'.

**Reviewer 1:** page 2, lines 9-10: this is a misleading statement not supported by the inappropriate reference (which is also quite old). it isn't necessarily the case that recovering or planted or re-planted forest has lower potential biomass many plantations or managed forests have higher growth rates than unmanaged forest, and in time could easily match or outgrow unmanaged biomass levels furthermore, the reference is about available land for afforestation, and does not compare afforested stands with corresponding primary stands. there are several recent papers that estimate carbon sequestration potential of forests. here is one example: griscom et al 2017: natural climate solutions in PNAS, oct 31 2017 vol 114 no 44 11645-11650

**Response:** We have revised these line as "*Afforestation/reforestation, in contrast, recovers forest which accumulates carbon but sequestration potential are constrained by water and nutrient availability (Smith and Torn, 2013)*", and the reference discusses ecological limit on sequestration potential of afforestation/reforestation.

**Reviewer 1:** page 2, lines 18-22: awkward sentence that is difficult to read. split it up and bring the examples out of the parentheses

**Response:** It has been revised as "*For this purpose, LULCC reconstructions enter Earth System Models (ESMs) (Lawrence et al., 2016), Dynamic Global Vegetation Models (DGVMs) (Le Quéré et al., 2018) and bookkeeping models (Hansis et al., 2015) to quantify biogeochemical and biophysical impacts of historical land-use change …*".

**Reviewer 1:** page 5, lines 22-23: are there other factors not taken into consideration? is the climatology constant or does it vary with time? be specific here - state exactly what factors are or are not included

**Response:** The climatology is produced by averaging temperature and precipitation of MSTMIP 1901 and 2000 and remained as constant over the spin up period. We have reorganized these lines.

**Reviewer 1:** page 5, lines 26-29: what tree density or how much tree canopy cover will give 2 KgC/m^2? what defines potential forest area in the two comparison studies?

**Response:** It is difficult to link biomass density to tree density as their relationship may strongly vary with tree species and also locations, and there are many different definitions of forest in the literature. The threshold value of 2 kg C/$m_2$ potential biomass was used for consistency with prior studies and GLM2/LUH2 (see references below).

*Hurtt GC, Pacala SW, Moorcroft PR, Caspersen J, Shevliakova E, Houghton R, Moore B (2002) Projecting the Future of the US Carbon Sink. Proceedings of the National Academy of Sciences of the United States (PNAS)/ 99(3): 1389-1394.*

*Hurtt GC, Frolking S, Fearon MG, Moore B, Shevliakova E, Malyshev S, Pacala SW, Houghton RA (2006), The underpinnings of land-use history: three centuries of global gridded land-use transitions, wood harvest activity, and resulting secondary lands. Global Change Biol 12:1208–1229*

*G. C. Hurtt, L. P. Chini, S. Frolking, R. A. Betts, J. Feddema, G. Fischer, J. P. Fisk, K. Hibbard, R. A. Houghton, A. Janetos, C. D. Jones, G. Kindermann, T. Kinoshita, Kees Klein Goldewijk, K. Riahi, E. Shevliakova, S. Smith, E. Stehfest, A. Thomson, P. Thornton, D. P. van Vuuren, Y. P. Wang (2011) Harmonization of land-use scenarios for the period 1500–2100: 600 years of global gridded annual land-use transitions, wood harvest, and resulting secondary lands. Climatic Change 109:117–161*

**Reviewer 1:** page 6, lines 5-28: These equations are incomplete, and therefore confusing. Transitions from types 5-8 to 5-8 are not included for f-gained. As such we do not see that these types of gains account for the f/l ratios of the losing types 5-8. Transitions from types 1-4 to 1-4 are not included for losses or gains. Doesn't harvest move primary to secondary land? Are there any gamma factors for these transitions? Also, wouldn't abandoned ag land move to secondary first, then from secondary to primary? For this path from ag to secondary to primary, how is the abandoned ag vegetation fraction tracked over time?

**Response:** As the example of forest-pasture transition (the $2_{nd}$ paragraph of section 2.3) explains, vegetation remaining in pasture will be cleared when the land is further changed to non-primary and non-secondary land. Similarly, vegetation remaining in cropland, rangeland or urban will also be cleared when the land is changed to cropland, pasture, rangeland and urban. Therefore, vegetation fraction could only be gained from land use changes from type 1-4. This is why $f^{gained}(i, t)$ in Eq.2 only includes land use changes from types 1-4. To avoid confusion, we clarify this before Eq.2.

Wood harvest from primary land does move vegetation to secondary, and there is no gamma factor for it. We have fixed the Eq. 5 and 6. Note that vegetation in cropland, pasture or rangeland could be back only to secondary land, there Eq.6 only apply to forested and non-forested secondary land. We have clarified this at the paragraph before Eq.4.

Besides, regarding the path you gave, the first part from abandoned ag to secondary is tracked in the Eq. 3 and 6, but the rest part that from secondary to primary is invalid in LUH2 according the definition of primary land in LUH2. Specifically, primary will not be reconverted once any land-use changes occur, thus your path will stop as secondary land.

**Reviewer 1:** Furthermore, there are corresponding equations for l(i,t+1), correct? But they are different because they track the land use transitions upon which the vegetation changes (without the gamma parameter that alters vegetation cover). This distinction and relationship between the two needs to be made clear.

**Response:** $l(i, t)$ is land-use fraction from LUH2 for type $i$ at time $t$. This is no need to track land use fractions as LUH2 provides time series for them. Changes are made at the paragraph after Eq.3.

**Reviewer 1:** page 6, lines 20-21: f(i,t) could also be equal to l(i,t)

**Response:** Yes, we have revised it as "*this fraction is larger than or equal to its vegetation fraction $f(i, t)$.*"

**Reviewer 1:** page 6, lines 22-23: "these" is confusing. maybe delete "these" and make "land-use" at the end plural.

**Response:** We have rephased the whole paragraph that describes how vegetation in primary and secondary land is tracked.

**Reviewer 1:** page 7, line 21: considering annual creation and discard of wood products and decay in landfills, these emissions can grow to be quite large over time.

**Response:** We agree that emissions from wood product could be very large. However, we think using biomass change as proxy of land-use change emissions in this manuscript has already included the emissions you mentioned, since wood products accumulate carbon from harvest, deforestation and other activities, and this accumulation could be reflected in vegetation carbon changes.

**Reviewer 1:** page 7, lines 22-32: This does not appear to account for different areas of secondary land within a grid cell with different ages. Similarly to my comment above, if cropland is abandoned and has half natural secondary veg and half crop veg, there can be at least three

different ages of secondary land in the cell. The number of different ages will grow with each transition. How is this dealt with? When does secondary become primary land again? Does primary land always have biomass density of B0?

**Response:** GLM2 does not account explicitly for the complete age distribution within secondary lands. When a fraction of secondary is newly created either from abandoned cropland or from harvested primary land, the mean age of the secondary land will be re-computed, and then it increases year by year to track biomass growth. Besides, primary land always has $B_0$, and once land use change occurs, it will never come back to primary land because of the primary land definition. We added description at the 2nd paragraph of section 2.4 to clarify the computation of mean age.

**Reviewer 1:** page 8, line 25: do you expect a 30% tree cover threshold to correspond with GLM's 2 KgC.m^2 threshold? Also, are the >30% pixel areas used for the values in table 2, or are the values in table 2 pre-threshold tree cover fractions that were calculated?

**Response:** Threshold value of 30% is discussed in Sexton et al., 2016 and its resulting forest area is comparible to FAO report. This value was used to convert tree cover to binary maps of forest and non-forest at 1km resolution, then we counted the forest area based on the 1km binary maps and reported global forest area of satellite-based dataset in Table 2. We have clarified this at the 2nd paragraph of section 2.5.

**Reviewer 1:** page 10, lines 4-28: What numbers are you comparing here and in figures 3-4 and table 4? Are the six-dataset numbers the tree cover or the forest with cover >30%? What does GLM's biomass threshold represent in relation to these metrics? Also, there are a couple of other tree cover estimates that are much lower: see meiyappan and jain 2012 frontiers in earth science 6(2):122 and Li et al 2018 earth system science data, 10:219.

**Response:** Only forest area/cover with a tree-cover threshold >30% of six satellite-based datasets are used in section 3.2 "Forest cover evaluation". We have clarified what metrics are used for evaluation at the last paragraph of section 2.5.

The lower global forest area estimates of meiyappan and jain 2012 may due to definition of forest, as they state in the second paragraph of section 4 "Comparisons with other studies". Moreover, the values of their test-case in the Table 5 is very comparable to the Table 2 of our manuscript.

The inclusion of six datasets as diagnostics in our manuscript is to avoid biased evaluation because of choosing a particular dataset. It will be very helpful to investigate the causes why satellite datasets report different forest cover, but it beyond the scope of our manuscript.

**Reviewer 1:** page 11: Can you make regional comparisons of LULCC emissions with info from the sources in table 3?

**Response:** As most of studies in Table 3 do not provide regional or gridded estimates, it is difficult to compare the regional LULCC emissions. However, we do compare the emissions from pasture and rangeland expansion for Rule 1-4 at different regions in Figure 6.

**Reviewer 1:** page 11, lines16-25: It would be helpful to see regional comparisons of carbon stock here.

**Response:** we have added Figure S4 for comparing carbon density at different latitudinal bands.

**Reviewer 1:** page 12, lines 11-13: up to this point you report that rules 1-3 have the same forest area dynamics, which makes sense because forest is always cleared in all three rules. but here you state that there are differences between rules 1 and 2 for forest area. please clarify why this is the case, and if they are different it needs to be noted early on, even if the differences are negligible.

**Response:** Rule 1,2, 3 should have the same forest area and only differ in carbon density, we have reorganized the results, discussion and conclusions section.

**Reviewer 1:** page 12, lines18-30: This is repetitive of your methods. It would be more useful to have discussion of the alternative rules, which you include in your figures/tables, but do not comment on at all.

**Response:** We have reorganized the discussion and conclusions, also analyzed Rules 5-9 at results section.

**Reviewer 1:** Figure 2 Only the results for rules 1-4 are shown, not all 9 as in the caption

**Response:** Right, we have fixed it.

**Reviewer 1:** Table 5 Do you mean compared with studies from table 3?

**Response:** Yes, it should be 'Table 3'. Change made.

**Reviewer 1:** Figures 5 and 6 I don't see the black dashed line. is it hidden?

**Response:** As Rules 1 and 3 have same treatment for land-use changes from primary and secondary land to managed pasture, the resulting emissions are very closed to each other, thus back dashed line almost completely overlapped blue dashed line. We have recreated Figures 5 and 6 by splitting each rule into individual plot.

**Reviewer 1:** Figure 8 I suggest plotting them in rule # order - rules 1 and 2 appear to be switched

**Response:** Change made.

---

## Author Comment (AC2) · 5 Dec 2019

Responses to reviews for manuscript **Global Transition Rules for Translating Land-use Change (LUH2) To Land-cover Change for CMIP6 using GLM2**

**To Reviewer #2 Eddy Robertson:**

**Reviewer 2:** The paper provides a useful assessment of how the choice of implementation of landuse data can affect the simulation of tree cover and carbon emissions. A more detailed description of the model used is needed to allow better interpretation of the results. A clearer justification of the choice of optimal transition rule is needed. Key uncertainties in the model simulation need to be discussed, in order to assess whether the results show the optimal transition rule for all models or simply for GLM2. I have also suggested a few minor corrections.

**Response:** Thank very much for your time on reviewing our paper. Your comments are very helpful for us to improve the paper. We have carefully addressed your comments, please see the detailed responses below. Note that 'transition rule' has been renamed as 'translation rule' throughout the manuscript to avoid confusion with LUH2 land-use transitions.

**Reviewer 2:** P5, L21: Please provide more detail about the Miami model and the simulation of carbon stocks.
– What inputs does the Miami model use? Does it use climate data? What is the MSTMIP climatology?
– Is the Miami model a process-based model or a statistical model?
– What time period is being simulated? What period does the MSTMIP climatology cover? Does the model use CO2 concentration from the year 850, or is CO2 concentration simply not a factor the model can consider?

**Response:** The GLM2 estimate carbon stocks and fluxes using a statistical model which take temperature and precipitation into account. The input is from MSTMIP by averaging temperature and precipitation during 1901-2000. Since this model does not take $CO_2$ into consideration, $CO_2$ concentration is not used at all. We have reorganized these lines and clarified these questions at the first paragraph of section 2.3.

**Reviewer 2:** Discussion: The paper should comment on the significance of uncertainty in the map of potential carbon stocks, for example if the global total potential carbon stock were only 557 Pg C, do you think a different transition rule would be optimal?
– Related to the above point, is whether different rules are best for different ESM, because they will simulate different potential carbon stocks. Please comment.

**Response:** We added a discussion of this issue to the Discussion. Briefly, our goal was to provide a reference set of translation rules for this reference land-use dataset. However, we note that it is possible to obtain different results with different models using the same translation rules. The CMIP6 Land Use Model Inter-comparison Project (LUMIP) has organized an inter-comparison of model results using this forcing dataset (Lawrence et al 2016).

**Reviewer 2:** P5, L26 and discussion: Why was 2 kg C/m2 chosen to define a forest? Is using 2.2 kg C/m2 equally justifiable and if so, might this lead to a different transitional rule being optimal?

**Response:** The threshold value of 2 kg C/m$_2$ potential biomass was used for consistency with prior studies and GLM2/LUH2 (see references below).

*Hurtt GC, Pacala SW, Moorcroft PR, Caspersen J, Shevliakova E, Houghton R, Moore B (2002) Projecting the Future of the US Carbon Sink. Proceedings of the National Academy of Sciences of the United States (PNAS)/ 99(3): 1389-1394.*

*Hurtt GC, Frolking S, Fearon MG, Moore B, Shevliakova E, Malyshev S, Pacala SW, Houghton RA (2006), The underpinnings of land-use history: three centuries of global gridded land-use transitions, wood harvest activity, and resulting secondary lands. Global Change Biol 12:1208–1229*

*G. C. Hurtt, L. P. Chini, S. Frolking, R. A. Betts, J. Feddema, G. Fischer, J. P. Fisk, K. Hibbard, R. A. Houghton, A. Janetos, C. D. Jones, G. Kindermann, T. Kinoshita, Kees Klein Goldewijk, K. Riahi, E. Shevliakova, S. Smith, E. Stehfest, A. Thomson, P. Thornton, D. P. van Vuuren, Y. P. Wang (2011) Harmonization of land-use scenarios for the period 1500–2100: 600 years of global gridded annual land-use transitions, wood harvest, and resulting secondary lands. Climatic Change 109:117–161*

**Reviewer 2:** P10, L27 and figure 4: Disagreement with the average satellite-based forest cover does not mean that the model is not consistent with (i.e. within the range of) the ensemble of satellite-based forest cover. Please can you at least mention this possibility. You could account for this uncertainty, perhaps by adding an uncertainty bar to figure 4 showing the errors relative to the TCCF and GLC2000 datasets.

**Response:** Good suggestion. We have recreated the Figure 4 by comparing rule estimates to each of the six datasets as well as to the average satellite-based forest cover. The new Figure 4 still suggests Rules 1, 2, 3 outperform Rule 4 in terms of gridded forest cover as long as the same satellite-based dataset is used as the reference.

**Reviewer 2:** Figures 5, 6 and 9: Please add results from rule 4 to figures 5, 6 and 9. I don't know why you have stopped considering rule 4.

**Response:** We stopped considering Rule 4 because it has relatively large bias than Rules 1-3 in forest cover and vegetation carbon estimates. However, we have included Rule 4 again in our recreated Figure 5 and 6.

**Reviewer 2:** Discussion: More justification is needed for you choice of optimal rule. I agree that, all else being equal, using rule 1 is best because it is consistent with HYDE 3.2. It's fair to exclude rule 4 because it produces too much tree cover (table 4) and compared to the other rules it has twice the carbon stock error in the tropics (Figure 8). However, in the discussion you need to more clearly state why rule 4 is excluded. I don't know why rule 3 has been excluded, please can you justify this choice?

**Response:** We have added detailed justification at the results section.

**Reviewer 2:** Discussion: Will the choice of rule matter less in future simulations? Is there less range-land expansion in future?

**Response:** We believe discussion of impacts of rule choices is very helpful, however our experiments and diagnostics really limit this kind of analysis, therefore we only state that this study only aim to propose a recommended rule for translation of historical land-use change to land-cover change in the introduction.

**Reviewer 2:** P2, L18-22: This sentence needs to be made clearer. I also think that you are underselling the importance of LUH2, it is not only used in land-use specific model simulations, it is a key input to the DECK and historical simulations as well as for future projections (scenarioMIP). Additionally, you could mention that it is used to simulate the biophysical effects of land-use change as well as the biogeochemical effects.

**Response:** We have rephased these lines to appropriately credit the significance and importance of the LUH2 for the reasons you mention. The revised sentence is "*Quantification of historical Land-Use and Land-Cover Change (LULCC) is important because it serves as the basis for examining the role of human activities in the global carbon budget and the resulting impacts to Earth's climate system. For this purpose, LULCC reconstructions enter Earth System Models (ESMs) (Lawrence et al., 2016), Dynamic Global Vegetation Models (DGVMs) (Le Quéré et al., 2018) and bookkeeping models (Hansis et al., 2015) to quantify biogeochemical and biophysical impacts of historical land-use change as part of historical simulates (DECK and CMIP6 historical simulations), future projections (scenarioMIP), impacts studies (ISIMIP), paleoclimate studies (PMIP), land-use specific simulations (LUMIP), and biodiversity studies (IPBES)*"

**Reviewer 2:** P3, L11-15: This sentence needs to be made clearer.

**Response:** We have reorganized this part by removing some irrelevant description. The revised sentence is "However, explicit suggestions for land-cover and carbon stock modifications resulting from these new defined land-use types are not yet provided, but are crucial for the translation of land-use change to land-cover change within ESMs or DGVMs".

**Reviewer 2:** P8, L1: What is the "wood fraction?"

**Response:** The fraction of NPP is allocated to cumulate stem and branch biomass annually. We have added explanation for it.

**Reviewer 2:** P13, L4: Please remove reference to the "Miami-LU model" and replace with "Miami model" or "GLM2," as appropriate.

**Response:** We have replaced 'Miami-LU' by 'GLM2'.

**Reviewer 2:** Caption of Table 5: Should refer to table 3 not table 4?

**Response:** Yes, it should be 'Table 3'. Change made.

---

## Author Comment (AC3) · 5 Dec 2019

Responses to reviews for manuscript **Global Transition Rules for Translating Land-use Change (LUH2) To Land-cover Change for CMIP6 using GLM2**

**To Reviewer #3:**

**Reviewer 3:** The manuscript entitled Global Transition Rules for Translating Land-use Change (LUH2) To Land-cover Change for CMIP6 using GLM2 by Ma et al. aims at recommending a global transition rule for translating LUH2 land-use forcing into land-cover changes in CMIP6 models. The authors simulate land-use induced land-cover changes based on a set of translation rules using the GLM2 model and prescribed LUH2 land-use transitions. Subsequently, emerging present-day forest cover, biomass density, and LUC emissions are evaluated against published estimates. The authors conclude by recommending a rule where all vegetation is cleared upon cropland/pasture expansion and only forest vegetation is cleared upon rangeland expansion.

The paper is technically correct, well written, timely in providing recommendations how to translate land-use forcing into land-cover changes for CMIP6 and the content is generally suitable for publication in GMD. I am wondering, however, if the conceptual design of the study is valid in the context of the large uncertainties that exist in the reconstruction of historical land use. Likewise, I do not think that the main conclusion ('optimal' rule being rule 1) is supported by the results presented. The wording ('optimal transition rule', 'accurate quantification', […]) is not suitable in the context of large uncertainties. Rather than claiming an 'optimal' transition rule, I suggest a framing towards recommending a 'consistent' translation rule for implementation of land use in ESMs/DGVMs. A consistent treatment in ESMs/DGVMs would eliminate added uncertainty and complexity from different treatment in each model. If this is 'optimal' (or and 'optimal' global rule even exists), is difficult to judge with available data and not shown by the contents of this manuscript. The reproducibility of the analysis presented is questionable, especially as neither the code of GLM2 (though stated in the code and data availability section) nor any documentation of the model is publicly available at the moment.

**Response:** Thanks for your comments which have helped us improve the manuscript. We have revised the manuscript including changing word of 'optimal' to 'recommended' and strengthening the justification of rule determination at result section by adding spatial analysis and regional comparisons. Note that 'transition rule' has been renamed as 'translation rule' throughout the manuscript to avoid confusion with LUH2 land-use transitions.

Regarding the comment of the conceptual design, we agree that large uncertainties exist in current land-use modelling products, and it is difficult or even impossible to propose globally 'optimal' rule to translate land-use change to land-cover change regardless of the land-use dataset. Therefore, we have modified the last paragraph of introduction section to clarify the goal that recommended through our evaluation only guide the implementation of LUH2 (historical part) in CMIP6. Besides, we agree with your point that "globally consistent rule could eliminate

added uncertainty and complexity from different treatment in each model", we have added this at end of the second last paragraph of introduction section.

**Reviewer 3:** Conceptual design. One of the main (implicit) assumptions in the manuscript is that the land-use transitions from LUH2 are 'correct', although these transitions are affected by large uncertainties and necessarily based on many assumptions. Given also the presented results that show simulated forest cover (and mostly also carbon emissions) in the range of previously published results for rules 1-4 (and partly even for the 'analytical' rules), I am wondering how valid any conclusions drawn regarding an 'optimal' translation rule can be. Without evaluation if any of the reference estimates is 'better' than others, I cannot see why Rule 1 is more 'optimal' than Rule 2, 3, or 4 (as long as they are all within the range). Moreover, if we can see only this small discrepancy already with assuming LUH2 data as 'correct', how do the authors think, would this evolve, if accounted for uncertainties in the land-use reconstruction? For example, how would the LUH2 high and low estimates change the results? How would prioritizing another land-use type in the allocation of land-use transitions in GLM2 (if this still exists like in GLM) change the results? And in conclusion: Does it even matter which of the rules is applied in an ESM/DGVM given these probably much larger effects from the mentioned uncertainties (besides of being consistent across CMIP6 models)?

**Response2:** Given the role of LUH2 as required forcing dataset, our goal was to determine the best translation rules given these data to inform and standardize the use in future modeling studies. To address uncertainty, we now include estimates of uncertainty in key reference datasets. We show that given this uncertainty, it is not technically possible to differentiate performance between some of the possible alternative rule choices. However, here we do confirm that Rule 1 performs among the best through these analyses consistent with the HYDE recommendation, and therefore have increased the confidence in recommending its standard usage. While LUH2 does provide a historical high-low, we focus here on the reference dataset only due to its required usage in model forcing. The Land-Use Model Inter-comparison Project (LUMIP) is organized to compare model performance using this forcing dataset.

**Reviewer 3:** Wording/Framing. Closely related to the comment above, I do not think that the wording in the manuscript is appropriate at many instances. The authors should avoid terms like 'accurate', 'optimal', etc. in the context of historical land-use change and its translation to land-cover change. As the authors state correctly in their discussions, globally valid transition rules probably do not exist and I would encourage the authors to rather emphasize the underlying uncertainties than trying to hide them behind strong words. Instead of claiming to derive 'optimal' rule(s), it would be more useful to recommend a 'reasonable rule' that should be used consistently across CMIP6 models.

**Response:** Very good suggestion. We have removed these adjectives where possible.

**Reviewer 3:** Conclusions. The main conclusion (=recommending rule 1) is not supported by the results presented. For all of the proposed diagnostics (forest cover, biomass density, carbon

emissions), all of the rules (1-4) are within the range of the diagnostics (sometimes even some of the analytical rules which are supposed to be idealized/unrealistic), and all of them are far from 'accurately' translating land-use change to land-cover change as the authors claim. The fact that on average one of the rules is closer to an averaged reference map does not provide justification that one of the four rules is superior over the others. The authors need to provide more justification why they recommend rule 1 and/or 2 based on the results presented here. Personally, I do not think this is possible without either defining a 'best/most suitable' reference map (e.g., based on GLM2 forest definition) or extended (spatial) analysis to identify regional characteristics of the different rules. Generally, the discussion/conclusion section needs to be strengthened, as in its current form it mainly repeats methods/results instead of discussing the findings of the analysis.

**Response:** We largely agree and have re-framed the analysis in response to these concerns. In the revised manuscript, we compare performance to each rule to each reference map as opposed to the average. We also include a treatment of uncertainty in the reference maps.  In the end we now show its difficult to differentiate between Rules 1-3 and recommend Rule 1 for both its relatively good performance and underlying prior recommendation from the HYDE.

**Reviewer 3:** General. Please check the whole manuscript for missing whitespaces in front of references and within references.

**Response:** Changes made.

**Reviewer 3:** P1 L15 'accurately' does not seem to be an appropriate wording given the large uncertainties both in climate and land-use modeling. Please remove.

**Response:** We have rephased this sentence to remove the word 'accurately'.

**Reviewer 3:** P1 L16 I would suggest to use 'land-cover change time series' only, i.e. remove the 'land-cover'.

**Response:** We have rephased it as "*climate models need consistent land-cover change time-series at a global scale*".

**Reviewer 3:** P1 L18-21 Please include that GLM2 was used for the simulations already here.

**Response:** They are changed to "*Building upon the latest Land Use Harmonization dataset (LUH2), land-cover dynamics, particularly in forest cover and carbon stock, were simulated based on each rule from 850 to 2015 globally by Global Land use Model 2 (GLM2) at quarter degree spatial resolution.*".

**Reviewer 3:** P1 L23-25 I think 'optimal transition rule' is not the correct wording. This sentence is also quite complicated. I would suggest to rephrase, emphasizing that within GLM2 the

mentioned rule turned out to perform best. The wording here indicates that this rule is 'optimal' irrespective of the model used, which is not supported (and probably also not intended) by the results of the manuscript.

**Response:** We have reorganized this sentence as "*Examinations at global, country, and grid scales indicate that the recommended translation rule for CMIP6 models is 1) completely clear vegetation in land-use ...*".

**Reviewer 3:** P1 L26-28 I am wondering if mentioning the detailed forest area is required, while not referring to the carbon density and LUC emissions at all?

**Response:** We have added results for carbon density there as "*According to this rule, contemporary global forest area is estimated to be 37.42 $10_6$ km2, and forest area estimates at global and country scales both stay within the range derived from remote sensing products. Likewise, the estimated carbon stock is in close agreement with reference biomass datasets, particularly over regions with 50% forest cover. This rule also mitigates the anomalously high carbon emissions from land-use change observed in previous studies in the 1950s*".

**Reviewer 3:** P2 L5-7 I think it is not correct to present this statement as a fact. The numbers are based on a historical LU reconstruction, i.e. model results. Please rephrase to, e.g., 'Model results show […]' or 'It has been estimated, […]'.

**Response:** We have rephased it as "*It has been estimated that, during the past 300 years, >50% of the land surface has been affected by human land-use activities, >25% of forest has been permanently cleared, and 10-44 $10_6$ km2 of land are recovering from previous human land-use disturbances*".

**Reviewer 3:** P2 L7 Include 'amongst others' as there are many more impacts of LUC and land management on the carbon cycle than deforestation, afforestation, and wood harvest.

**Response:** P2 L8-10 has modified as "*Impacts on the carbon cycle result from several processes among others: deforestation removes natural forest and ...*"

**Reviewer 3:** P2 L12-13 Same as above. Please highlight that this is also an (uncertain) model result, e.g. by including the uncertainty ranges.

**Response:** We have rephased this as "*Cumulatively, models estimate that land-use land land-use change have contributed to a net flux 190±75 Pg C to the atmosphere during 1870-2017 (Le Quéré et al., 2018)*".

**Reviewer 3:** P2 L13-14 What are 'these emissions'? Please rephrase in a way that the reference to land-use emissions becomes clear.

**Response:** We have rephased it as "*While emissions from land-use and land-use change only account for 10% of current anthropogenic carbon emissions, they were a dominant contributor to increasing the atmospheric $CO_2$ above pre-industrial levels before 1920*"

**Reviewer 3:** P2 L13-15 The numbers presented here are probably all derived by using LUH(2) as land-use forcing. Thus, I think it would perfectly fit the storyline to add a sentence that explains that exactly due to these uncertainties a 'better' translation between land use and land cover is required. Otherwise, one may ask, why we would need all these transition rules, if we already know about historical land-use impact.

**Response:** We have added uncertain range at the end of 1st paragraph of introduction and pointed out at the next paragraph that a globally consistent translation rule is required for ESMs and DGVMs.

**Reviewer 3:** P2 L18-22 What about just saying LULCC reconstructions enter Earth System Models (ESMs) (e.g., Lawrence et al., 2016), Dynamic Global Vegetation Models (DGVMs) (e.g., Le Quéré et al., 2018), and bookkeeping models (Hansis et al., 2015; Houghton and Nassikas, 2017) to quantify biogeochemical and biophysical impacts of historical land-use change.' I don't think the details about models and MIPs is required here.

**Response:** We have changed these lines as "*LULCC reconstructions enter Earth System Models (ESMs) (Lawrence et al., 2016), Dynamic Global Vegetation Models (DGVMs) (Le Quéré et al., 2018) and bookkeeping models (Hansis et al., 2015) to quantify biogeochemical and biophysical impacts of historical land-use change as part of historical simulates (DECK and CMIP6 historical simulations), future projections (scenarioMIP), impacts studies (ISIMIP), paleoclimate studies (PMIP), land-use specific simulations (LUMIP), and biodiversity studies (IPBES)*".

**Reviewer 3:** P2 L18-22 Remove '(e.g., HYDE, SAGE)'. Replace 'Goldewijk et al., 2017' by 'Klein Goldewijk et al., 2017'. Add Ramankutty and Foley, 1999 and Pongratz et al., 2008.'

Ramankutty, N. and Foley, J. A.: Estimating historical changes in global land cover: Croplands from 1700 to 1992, Global Biogeochem. Cycles, 13(4), 997–1027, doi:10.1029/1999GB900046, 1999.

Pongratz, J., Reick, C., Raddatz, T. and Claussen, M.: A reconstruction of global agricultural areas and land cover for the last millennium, Global Biogeochem. Cycles, 22(3), GB3018, doi:10.1029/2007GB003153, 2008.

**Response:** These lines are changed to "*Considerable efforts have been devoted to modelling historical land-use states (Goldewijk et al., 2017; Kaplan et al., 2009; Pongratz et al., 2008; Ramankutty and Foley, 1999) and …*".

**Reviewer 3:** P2 L24-26 The manuscript is about historical land use, i.e. the harmonization with future LULCC seems to be an irrelevant information here.

**Response:** The lines are changed to "*In particular, the recent Land-Use Harmonization 2 (LUH2) dataset (Hurtt et al., 2017) has been developed to provide global gridded land-use states and transitions in a consistent format for use in ESMs as part of CMIP6 experiments*"

**Reviewer 3:** P2 L28-33 Remove the reference 'Shevliakova et al., 2013' between the sentences and only put it in the end of the paragraph.

**Response:** We have moved the citation to the end of the paragraph.

**Reviewer 3:** P3 L1-3 I would not agree that there is 'a lack of explicit global rules'. As the authors show later on, it is relatively easy to come up with some. I would rather argue that there is no consistency/agreement on which rule to apply. Apart from that, in this context it is also worth to mention that such 'global transition rules' probably do not exist at all [see, e.g., Prestele et al. 2017].

Prestele, R., Arneth, A., Bondeau, A., de Noblet-Ducoudré, N., Pugh, T. A. M., Sitch, S., Stehfest, E. and Verburg, P. H.: Current challenges of implementing anthropogenic land-use and landcover change in models contributing to climate change assessments, Earth Syst. Dyn., 8(2), 369–386, doi:10.5194/esd-8-369-2017, 2017.

**Response:** We agree that there is no agreement on which rule should be used, thus we have changed 'a lack of explicit global rules' to "a globally consistent rule".

**Reviewer 3:** P3 L5 … 'and the location where a land-use change happens'.

**Response:** It has been changed as "*the degree of land-cover alteration varies with the types of land-use changes and the location where a land-use change happens*".

**Reviewer 3:** P3 L5-7 While this statement sounds very intuitive, I wonder if there is any literature supporting these tendencies?

**Response:** We agree that the potential activity is intuitive, but are also not aware of specific literature on it.

**Reviewer 3:** P3 L11-15 Complicated sentence. Please shorten. In my opinion, everything after '…not yet provided' is not necessarily required. What about joining with the following sentence

instead? Isn't it exactly what the authors are aiming at: providing recommendations how to treat these 'new land-use types' in the translation?

Suggestion: 'However, explicit suggestions for land-cover and carbon stock modifications resulting from these new defined land-use types are not yet provided, but are crucial for the translation of land-use change to land-cover change within ESMs or DGVMs. An inconsistent translation will potentially produce very different land-cover dynamics, which will impact the land surface biophysical and biochemical processes.'

**Response:** Great suggestions. these lines are changed as "*However, explicit suggestions for land-cover and carbon stock modifications resulting from these new defined land-use types are not yet provided, but are crucial for the translation of land-use change to land-cover change within ESMs or DGVMs. An inconsistent land-cover translation of these land-use products within an ESM or DGVM will potentially produce very different land-cover dynamics, which will impact the land surface biophysical and biochemical processes.*"

**Reviewer 3:** P3 L18 I would not agree that the approach presented here will reduce any uncertainty. It rather can provide recommendations for consistent treatment across models, if the 'optimal' rule is adapted by the CMIP6 models. But this does not allow any conclusions how uncertainty will be affected.

**Response:** these lines have been revised as "*To recommend a global translation rule for translating historical land-use changes for CMIP6 models, this study investigates the impacts of land-use change on land-cover by proposing several alternative sets of translation rules, which ae then integrated into the Global Land use Model 2 (GLM2) model (Hurtt et al., 2017, 2019) to simulate ...*"

**Reviewer 3:** P3 L22 Remove 'other' in front of 'independent'.

**Response:** change made.

**Reviewer 3:** P3 L22-25 Here, too, I recommend not using the term 'optimal'.

**Response:** We have changed 'optimal' to 'recommended'.

**Reviewer 3:** P4 L1-5 Is there a specific reason not to include the ESA CCI land cover for comparison? Remove the details about the comparison dataset here. They are all mentioned in section 2.5

**Response:** the ESA CCI land cover dataset has different land-cover classification schemes with the six satellite-based datasets, and therefore the legend translation (Table S1 from Song et al 2014) may be not applicable for ESA CCI. We have re-organized P4 L1-L5.

**Reviewer 3:** P4 L7-28 As the method section is already quite long, I would suggest to shorten here. I do not think there is a lot of added value to describe LUH2 in this detail for the purpose of the paper. I guess there will be an associated LUH2 publication soon, so it is probably enough here to just describe the key features that are relevant for the analysis in this manuscript.

**Response:** section 2.1 has been shortened.

**Reviewer 3:** P4 L11-13 This sentence doesn't seem to fit in the context here.

**Response:** It has been removed.

**Reviewer 3:** P4 L17 While 'data-driven' is probably not wrong, I think it is misleading as it implies that the constraints used in LUH2 are based on observations. However, to my knowledge most of the constraints are model outputs in some way (be it the HYDE reconstruction or models derived from remote sensing images, etc.). Therefore, I would recommend not to use 'data-driven' here.

**Response:** We have removed 'data-driven' and this revised sentence is "*The LUH2 dataset was generated with the GLM2 (Hurtt et al., 2017, 2019), which like its predecessors (Hurtt et al., 2006, 2011), estimates annual sub-grid-cell land-use states and transitions by including multiple constraints such as gridded patterns of historical land-use from the HYDE database (Goldewijk et al., 2017), ...*".

**Reviewer 3:** P4 L28 Where do the 2 kg C/m² come from? How do they relate to other forest definitions? Are there any references that could support this threshold? Some more information would be valuable for the reader.

**Response:** It is difficult to link biomass density to tree density as their relationship may strongly vary with tree species and also locations, and there are many different definitions of forest in the literature. The threshold value of 2 kg C/m2 potential biomass was used for consistency with prior studies and GLM2/LUH2 (see references below).

*Hurtt GC, Pacala SW, Moorcroft PR, Caspersen J, Shevliakova E, Houghton R, Moore B (2002) Projecting the Future of the US Carbon Sink. Proceedings of the National Academy of Sciences of the United States (PNAS)/ 99(3): 1389-1394.*

*Hurtt GC, Frolking S, Fearon MG, Moore B, Shevliakova E, Malyshev S, Pacala SW, Houghton RA (2006), The underpinnings of land-use history: three centuries of global gridded land-use transitions, wood harvest activity, and resulting secondary lands. Global Change Biol 12:1208–1229*

*G. C. Hurtt, L. P. Chini, S. Frolking, R. A. Betts, J. Feddema, G. Fischer, J. P. Fisk, K. Hibbard, R. A. Houghton, A. Janetos, C. D. Jones, G. Kindermann, T. Kinoshita, Kees Klein Goldewijk, K. Riahi, E. Shevliakova, S. Smith, E. Stehfest, A. Thomson, P. Thornton, D. P. van Vuuren, Y. P. Wang (2011) Harmonization of land-use scenarios for the period 1500–2100: 600 years of global gridded annual land-use transitions, wood harvest, and resulting secondary lands. Climatic Change 109:117–161*

**Reviewer 3:** P5 L1 What are the 'analytical purposes' of rules 5-9? For the rest of the manuscript they are mostly used to state that the results with these results are 'way off', but this is not very surprising given their idealized/unrealistic character. I would therefore recommend to leave them out, as they rather add confusion.

**Response:** Rules 5-9 are included for model experimental design completeness and for reference to the other cases. In the text, we clearly differentiate them as such.

**Reviewer 3:** P5 L8-12 I agree that the effect of spatial and temporal varying rules is beyond the scope of this study. However, these are very strong simplifications and it would be useful to get an indication of how including this variation would affect the results. Maybe the authors could look a bit more detailed into the country-level and gridded results for the different rules and diagnostics. Can there be seen any patterns, if one of the rules is 'more likely' in certain regions than in others? If this is not feasible, the authors should include more detailed elaboration how the results may be affected in the discussion section.

**Response:** We have added regional comparison of Rules 1-3 estimates of carbon density at Figure S5.

**Reviewer 3:** P5 L18-19 It sounds a bit 'circular' that the output of GLM2 (i.e., LUH2) is used as input into GLM2 for the analysis in this manuscript. Could the authors provide more explanation how this was implemented? Are these independent model runs?

**Response:** To better explain the model runs, we have rephased these lines as "*In this study, land-cover change is simulated by performing a modified GLM2 simulation in which the computed land-use transition rates (using the same methodology as LUH2) are supplemented with a set of translation rules (Table 1) to track forest cover change and carbon dynamics at 0.25° spatial resolution*"

**Reviewer 3:** P5 L23-24 More detail required regarding the model run (time period, etc.).

**Response:** We have updated the model info such as inputs, time period, environmental factors not considered at the first paragraph of section 2.3.

**Reviewer 3:** P5 L23-29 These are effectively 'results'. I would recommend to move to section 3.1.

**Response:** They have been moved to section 3.1.

**Reviewer 3:** P5 L31 ff. I do not know about the specifications of GLM2/LUH2, but in LUH1 [Hurtt et al. 2011], choices had to be made about starting date, priority for land-use transitions, wood harvest inclusion, etc. If this still exists for GLM2/LUH2, it would be useful to indicate here, which configuration of GLM2 was used to derive LUH2 to allow the reader to understand how the historical transition rates have been derived. In the discussion, a short evaluation of how changing these assumptions would change the results of the analysis, would help.

Hurtt, G. C., Chini, L. P., Frolking, S., Betts, R. A., Feddema, J., Fischer, G., Fisk, J. P., Hibbard, K., Houghton, R. A., Janetos, A., Jones, C. D., Kindermann, G., Kinoshita, T., Klein Goldewijk, K., Riahi, K., Shevliakova, E., Smith, S., Stehfest, E., Thomson, A., Thornton, P., van Vuuren, D. P. and Wang, Y. P.: Harmonization of land-use scenarios for the period 1500-2100: 600 years of global gridded annual land-use transitions, wood harvest, and resulting secondary lands, Clim. Change, 109(1), 117–161, doi:10.1007/s10584-011-0153-2, 2011.

**Response:** For the LUH1 dataset we performed a large sensitivity analysis in which we systematically varied model inputs and decisions. However, owing to the complexity and increased detail of LUH2 we have not performed a similar sensitivity study for this dataset. As such, there are now choices to be made by the user regarding the specifications of the dataset. Of course, this does not imply that there is no uncertainty in the various model factors and changing the inputs and model decisions would ultimately change the carbon accumulated in the land surface, but would not necessarily change the overall recommendation of which translation rule to use.

**Reviewer 3:** P7 L26 Not an expert in carbon impacts of land management. However, I am wondering if management of land necessarily means that there is no further accumulation of biomass in the remaining 'natural' vegetation?

**Response:** When natural vegetation remains on managed land there are a range of possibilities – the accumulation of carbon could decrease over time (if for example that vegetation is grazed), it could remain constant (if it is explicitly managed and with the intent to keep it), or it could grow with time (if it is allowed to expand spatially). In the absence of any other information we have chosen a simple, middle-of-the-road, assumption that the carbon in remaining vegetation is explicitly managed and remains constant. Furthermore, we have acknowledged the possible impact of the assumption at the results section that consistent underestimation of carbon stocks in Rules 1-3 may be related to this assumption.

**Reviewer 3:** P9 L20 Instead of only using an average 'smallest difference' for the gridded results, looking more into the spatial patterns would maybe help to derive stronger justification for recommending one of the suggested rules.

**Response:** We have added difference maps of forest cover and carbon density (Figure 2, S2 and S3), and zoom in some regions for detailed comparison (Figure S5).

**Reviewer 3:** P9 L29 'Higher forest cover' compared to what? The average reference map? Unclear. In addition, rather than presenting the three forest cover maps in Fig. 2, it would be more useful to show difference maps (rule 1-3 minus reference; rule 4 minus reference). This would facilitate the identification of differences between the maps.

**Response:** "higher" has been changed to "high", we just want to claim Rules 1-4 could reproduce the general pattern of forest cover such as where has more forest than other places. Furthermore, we have added the difference maps in Figure 2 for better comparison.

**Reviewer 3:** P10 L9-11 Here, the authors emphasize the uncertainty in the definition of 'forest', which cannot easily be resolved. Which definition used in one of the reference maps is closest to the forest definition used in GLM2 ($> 2$ kg C/m²). Using the 'closest' map to compare the GLM2 results to would probably give a better indication than having a huge range of 'reference' maps (where the range partly originates 'only' in definition issues).

**Response:** It is good to evaluate rules with multiple independent satellite dataset to reduce uncertainties originating from particular methods or sensors, but also difficult to visualize the comparison between reference maps and model estimates. Here, we still prefer to keep these six satellite datasets, but extend Figure 4 by including comparisons to all rule to each of the six datasets rather than the averaged dataset. The extended analysis still supports our conclusions that Rules 1, 2, 3 outperform Rule 4 in terms of gridded forest cover.

**Reviewer 3:** P10 L12-15 Rule 7 is within the range according to Fig. 3.

**Response:** These lines haven been updated to "*The forest cover based on Rules 6, 8 and 9 is beyond the range of the diagnostics, indicating that these rules underestimate the impacts of land-use change on land-cover and overestimate the global forest existing in the present day.*"

**Reviewer 3:** P10 L16-17 Rule 5 and 7, too (see Fig. 3).

**Response:** These lines have been modified to "*In contrast, Rules1-4, 5 and 7 produced estimates of global forest area within the range of diagnostics.*"

**Reviewer 3:** P10 L19-20 The statement is not wrong, but also the analytical rules 'locate' around 75% of global forest land in these eight countries. This cannot be used as a characteristic of distinction between the rules.

**Response:** The sentence "*Rules 1-4 also produce the same pattern of locating most forest land within these eight countries (Table 4).*" has been removed.

**Reviewer 3:** P10 L27-28 I am not sure, if the mean average delta compared to an average global forest map is a good metric here. The average reference map is a rather 'artificial' map, and not necessarily the most plausible one. Does the assessment of 'smallest' difference change, if compared to the reference maps individually? Rather than using averages, I think the authors should aim at identifying a reference map that corresponds most with the forest definition within GLM2 and compare to this map. Additionally, showing a map of the differences would also allow to identify if certain rules match the current situation better in particular regions. As several rules are within the range of published forest cover areas, this would allow a better justification for one certain rule and/or regional diversification.

**Response:** Good suggestion to determine a reference map for evaluation. However, it is difficult to find such one which correspond most with GLM definition, as GLM does not has forest cover map before applying the rules of the paper. Instead, we have extended the Figure 4 by comparing rule estimates to each of six satellite-based forest cover maps as well as to the averaged map. We also added difference maps in Figure 2 for better justification.

**Reviewer 3:** P10 L10-12 What happened to rule 4? From Table 5 it can be seen that it reduces the pasture anomaly, too. Why does it not appear any more in the text and Figs. 5/6?

**Response:** Emissions of Rule 4 have been added to Figure 5 and 6.

**Reviewer 3:** P10 L13 Is the difference of 1 Pg C really a 'significant' difference?

**Response:** We have rephased this sentence as "*Rule 2 reduces more anomalous emissions than Rule 1 (reduced 6 Pg C in Rule 1 and 7 Pg C in Rule 2),…*"

**Reviewer 3:** P11 L16-20 The differences in the average difference between model and reference are rather small across all rules (except for some of the analytical). Again, are the authors sure that this average difference is a suitable indicator (see also comment above reg. forest area)?

**Response:** The "small" difference across rules is what we anticipated for two reasons. First, Rules 1-3 have same treatment of land-use changes to cropland, but different treatment for those from non-forested land (primary and secondary) to managed pasture or rangeland. Second, the carbon density of non-forested land is also relatively small (usually below 2 kg C/m2 in GLM2). Therefore, the small difference in carbon density and emission estimates are expected. For better readability of Figure 8, we rescaled the y-axis, we can see Rules 1-3 outperforms Rule 4. And also, we have also calculated the averaged difference over different regions (i.e. four latitudinal bands in Figure S4) instead of averaging over globe.

**Reviewer 3:** P11 L22-25 In Fig. 9 hardly any difference can be seen for the three rules. How large are the differences between the individual rules and the reference? How do the authors conclude from this Fig. that rule 1 and 2 are closer than rule 3? What happened to rule 4?

**Response:** We have modified Figure 9 by calculating the relative difference in carbon stock comparing to two carbon density maps. Rule 4 is excluded as it shows large bias in forest cover in Brazil and also larger averaged difference in carbon density (Figure 3), Figure 9 compares carbon stock of Rules 1-3 and then determine which should be recommended.

**Reviewer 3:** P11 L28-30 Please see major comment 'Conclusions.'

**Response:** Please see the response to comment 'Conclusions' at P11.

**Reviewer 3:** P12 L1-5 These statements are not wrong, but only repeat parts of the results. Please remove.

**Response:** We have reorganized the discussion section, and these lines also have been removed.

**Reviewer 3:** P12 L7-8 How do the authors think that the results presented here can facilitate the reconstruction of historical land-cover change? Please elaborate.

**Response:** We think by combining the LUH2 land-*use* transitions with the suggested rules for land-*cover* changes that could occur during those land-use transitions, it would enable the reconstruction of land-cover changes over the entire historical period. We have re-organized the whole discussion.

**Reviewer 3:** P12 L10 ff. It is still not clear at this point why rule 1/2 are 'better' than rule 3/4. Additionally, I wonder what is the added value of the study, if one of the main conclusions is that rule 1 is 'better' than rule 2 due to assumptions taken in HYDE.

**Response:** Our analysis takes the suggested rule from the HYDE3.2 paper, and subjects that rule to a series of tests designed to determine if that rule is consistent with multiple datasets and assumptions. This analysis goes beyond that which is presented in the HYDE paper. Furthermore, this analysis is performed within a consistent modeling frame-work, and in particular, with a different underlying map of land-cover/biomass.

**Reviewer 3:** P12 L24-30 While not wrong, I think irrelevant here as it (1) mainly repeats what has been written in the methods section and (2) does not provide justification for one of the rules presented. The authors should aim at emphasizing the reasons why they recommend rule 1 to CMIP6 models.

**Response:** We have re-organized the discussion section and also explained why rule is recommended.

**Reviewer 3:** P13 L6 Not clear why the authors introduce a new model here.

**Response:** we have removed it.

**Reviewer 3:** P13 L14 I do not agree with this statement/conclusion. Several rules presented here lead to similar results (within the range of reference maps) and justification is missing, why one of the rule is better than another. The claim that an 'optimal' rule has been determined by the analysis is not supported by the results.

**Response:** We have changed 'optimal' as 'recommended' and added extra figures and explanation at results section to support the justification of rule recommendation.

**Reviewer 3:** Figure 1 Isn't Fig. 1(b) a binary map (forest/no-forest)? In this case the legend doesn't make sense.

**Response:** No, it is the forest cover map of which most of grid-cells have fractions close to 100% at year of 850.

**Reviewer 3:** Figure 2 Please add difference maps to facilitate the identification of differences between the maps. Only 4 rules (instead of 9 as mentioned in the caption) are shown.

**Response:** Caption is corrected, and also the difference maps are included.

**Reviewer 3:** Table 1 Are the 'analytical rules' required for the purpose of the manuscript?

**Response:** Inclusion of Rules 5-9 could be used to interpret individual impacts of cropland, managed pasture and rangeland expansion, also give baseline estimates of resulting forest cover and vegetation for ESMs/DGVMs if similar rules are implemented.

**Reviewer 3:** Figure 3 For the analytical rules the x-axis labels are not centered any more.

**Response:** Figure 3 recreated to address this concern.

**Reviewer 3:** Figures 5-6 Why is rule 4 omitted from these Figs.?

**Response:** Rule 4 is included in the recreated Figures 5 and 6.

**Reviewer 3:** Figure 7 Also here difference maps would help to guide the reader.

**Response:** The difference maps have been added in Figure S2

**Reviewer 3:** Figure 8 Switch Rule 1 – Rule 2 (x-axis).

**Response:** Figure 8 recreated.

**Reviewer 3:** Figure 9 Differences between the rules hardly can be seen. Maybe zooming in for different percentages would improve the readability?

**Response:** We have modified Figure 9 by adding two zooming-in subplots.

---

## Referee Report (RR1)

**General comments**

I appreciate that the authors reconsidered the wording throughout the manuscript and added analysis, now presenting a more nuanced discussion of their approach to test different rules for the translation between LUH2 land-use and land-cover for ESMs. However, their main conclusion (=recommendation of rule 1) is not covered by the results of the analysis and sufficient justification for the exclusive recommendation of this rule is still missing. The framing of the manuscript still indicates the opposite, e.g. by the following statement in the abstract:

'*Examinations at global, country, and grid scales indicate that the recommended translation rule for CMIP6 models is 1) completely clear vegetation in land-use changes from primary and secondary land (including both forested and non-forested) to cropland, urban land, and managed pasture; 2) completely clear vegetation in land-use changes from primary forest and/or secondary forest to rangeland; 3) keep vegetation in land-use changes from primary non-forest and/or secondary non-forest to rangeland. This confirms the translation rules suggested earlier in the HYDE dataset underlying LUH2.*'

(1) The examinations across scales do not exclusively indicate rule 1 (instead rules 2 and 3 are equally likely), which the authors also state in the manuscript and in the reply to the reviewers.

(2) The examinations do not confirm the translation rule suggested by HYDE. Instead, the earlier suggestion from HYDE is used as (the main) justification to pick rule 1 instead of rules 2 or 3.

One way out would be to be very clear about the fact that rule 1 is only recommended to achieve consistent implementation in future simulations (i.e., it would require to be a major point in the discussion and also in the abstract) and this recommendation is NOT a result of the analyses in this manuscript (as these show that with the same arguments also recommendation of rules 2 and 3 could be justified).

In this context, the manuscript would also benefit from a more critical discussion about the downsides of a consistent 'translation rule' (which is not necessarily supported by available data). In my opinion, it is reasonable to aim at a standardized translation between LUH transitions and ESM land cover. But such a standardization always comes at the cost of omitting uncertainties, instead of actually reducing them. If, for example, the '*added uncertainty of 43 PgC in CMIP6*' (as stated in the discussions) is avoided by implementing a consistent 'translation rule' this does not necessarily mean that the uncertainty is not there anymore; it might be just not depicted in the ESM results anymore. Only if the authors could show by their analysis that one rule performs significantly better than others, this would be an indication for actually 'reducing' uncertainties.

If the authors do not want to put more emphasis on the consistency aspect and/or highlight the limitations of their results (i.e., basically we do not know about the 'correct' rule), they would need to show with their analysis that rule 1 outperforms the other rules.

In sum, I think it is a useful study/analysis and worth to be published, but requires more nuance in the presentation of results, limitations, and derived conclusions.

Some of the comments from the previous review were poorly addressed (pages/lines refer to the original comments/manuscript).

| | |
|---|---|
| P3 L1-3 | The authors did not address the comment on the (non-)existence of 'global transition rules' |
| P3 L5-7 | What is the basis for this statement, if it's not supported by literature? Some previous analysis? I think without a reference it is a misleading statement. |
| P4 L1-5 | I am sure there are suitable legend translations for ESA CCI land cover as well and it's one of the most up-to-date datasets, but indeed it's not a critical issue. |
| P4 L28 | While I see that it is difficult to link biomass density to tree density as the authors state, I think it would be worth to give an indication which one of the forest definitions in the literature (and also the ones in the reference maps used for comparison) is closest to this 2 kgC/m² definition. This definition has the potential to affect the results and deserves some attention. |
| P5 L1 | The intention to include rules 5-9 is still not clear. Although it might be useful for test/sensitivity runs (also for the ESM community), I think it doesn't make sense to include them if the main purpose of the manuscript is to derive a realistic/recommended translation rule (where these rules by definition are not useful). In the results (incl. tables and figures) they are hardly revisited and rather add confusion to some of the results. In my opinion, the authors should decide to either include all rules in all results/tables/figures or stick to rules 1-4. To concentrate on a different set of rules at different sections of the results is confusing. |
| P10 L9-11 | I see the authors intention to include the whole range of currently available forest reference maps. However, it would be still useful to give an indication which one is closest to the GLM2 forest definition. If we would know, for example, that one of the products has a similar forest definition, this could increase the confidence/justification for one of the rules. |

**Minor comments**

Page/line numbers refer to the revised manuscript.

| | |
|---|---|
| P1 L30 | Reference biomass is also close for rule 2 and 3. |
| P1 L30 | Should it be: '[…] regions with forest cover larger than 50%'? |
| P2 L16 | As there is now already a carbon budget update, it might be good to use the latest values/reference. |
| | Friedlingstein, P. et al. 2019. Global Carbon Budget 2019. Earth Syst. Sci. Data 11, 1783–1838. https://doi.org/10.5194/essd-11-1783-2019 |
| P3 L7-9 | It is not only the lack of a globally consistent rule, but also the fact that the existence of such a global rule is very unlikely and a large simplification (see original comment P3 L1-3). |
| P3 L21-23 | But also obscures the uncertainty from the lack of process understanding and lack of dedicated spatially explicit treatment. |
| P3 L25 | 'which **are** then integrated' |
| P9 L20 | 'accounted for in bookkeeping **model** based studies' |
| P9 L29 | 'should be **close** to diagnostics' |
| P9 L31-33 | It's not 'other criteria, such as …', but the only one that is used in the end to identify the recommended rule. |
| P10 L17-19 | I don't understand what the authors intend to say here? |
| P11 L2-4 | Due to these large discrepancies it would be even more helpful to guide the reader with some information about which forest definition (of the reference maps) is closest to the GLM forest definition. (see original comments P4 L28; P10 L9-11). |
| P11 L17-18 | And are within the range for Brazil, US, Congo, Indonesia, Peru. |

| | |
|---|---|
| P12 L6-14 | All the realistic rules (1-4) reduce the pasture anomaly. Is this then just the difference between LUH1 and LUH2 or really a characteristic of the individual rules? |
| P13 L3-5 | On average and globally. The regional and gridded comparisons (Table 4, Supplements) indicate that this might not hold at the country and grid level. Misleading statement. |
| P13 L6-8 | It's actually hard to say if it is 'better' given all the uncertainties in these comparisons. |
| P13 L14-16 | Which is also true for rule 2 and 3. |
| P13 L23-25 | The uncertainty is not really reduced by implementing a consistent rule, as long as we do not know, which rule is 'correct'. It's just omitted from evaluation. |

---

## Author Response (ED1)

Responses to reviews for manuscript **Global Rules for Translating Land-use Change (LUH2) To Land-cover Change for CMIP6 using GLM2**

**To Reviewer #1:**

**Reviewer 1:** This is a revision of a previously reviewed manuscript. The authors have done a good job addressing the reviewer comments (which was not a trivial task), but there are still a few things that need clarification before publication. I don't have any major concerns. Please see the comments below for details.

**Response:** Thanks for all comments. We have made some clarifications accordingly. Please see our point-by-point response below.

**Reviewer 1:** page 3, lines 19-23: It isn't clear what "inconsistent land-cover translation" means here. Inconsistent with LUH? Inconsistent across ESMs/DGVMs?
It seems that the meaning here is leaning toward inconsistency across ESMs/DGVMs, but both inconsistencies are relevant. So I suggest clearly specifying both.
And "globally consistent" is also ambiguous. A rule that is global in spatial extent? Or a rule that is applied consistently by different folks around the globe? Again, it seems like the latter makes more sense here. That it is a global rule is different issue that also generates uncertainty.
Also, "eliminate added uncertainties" is an overly ambitious claim.
Maybe try "Consistent application of a specified rule for translating…could reduce uncertainties from translation inconsistencies in studying…"

**Response:** 'consistent' here means ESMs/DGVMs are suggested to use the same rule to translate the given land-use change dataset such as the LUH2. We agree with your suggestion and changed the lines to "*Therefore, consistent application of a specified rule for translating land-use products could reduce uncertainties from translation inconsistency in studying land-use effects through ESMs and DGVMs*".

**Reviewer 1:** page 4, lines 14-16: Please clarify the relationship to LUH2. These GLM2 runs generate and track the exact same LUH2 data as before, and the additional translational tracking does not affect the LUH2 land use transitions. The translation and tracking of vegetation carbon is an additional capacity.

**Response:** Very good point. We have added this clarification into the first paragraph of section 2.3. It is "*Note that the modified GLM2 still generate and track the exact same land-use transitions of the LUH2 and has the additional function to track associated land cover change in terms of forest cover and vegetation carbon.*"

**Reviewer 1:** page 5, lines 17-20 Does this mean that the constant spin-up climate is a 100-year average?
How were the stocks and fluxes calculated during the translation simulations?

Did the spin-up produce a spatial, but temporally static, look-up table for use by the simulations, or was it just for initial conditions?
Or are the simulations also driven by some form of static or time-varying climate that determines carbon fluxes and stocks?
It appears later on page 7 that the spinup contributes to parameters for eq. 7. This should be clarified here.

**Response:** Yes, the 100-year averaged temperature and precipitation are used and remain constant during the spin-up. They are spatially varied but temporally static. Besides, the GLM2 use them to estimate fluxes and stocks (NPP and B(t) at Eq.7), which is explained at the first paragraph of section 2.3. For better clarification, we have changed the description of climatology generation as "*The annual temperature and precipitation maps from MSTMIP were averaged over 1901 and 2000 to generate the spatially varied and temporally static climatological temperature and precipitation, which was then used to spin up the GLM2 globally at 0.25x0. 25° resolution for 500 years.*". Besides, we also added explanation at the paragraph after Eq.7 "*Note that $B_0$ and $NPP_0$ are estimated by a statistical model in GLM2 using climatological temperature and precipitation and are constant over simulation period from 850 to 2015.*"

**Reviewer 1:** page 5, line 30
Is the type of vegetation remaining in the land use categories (5-8) tracked? Or is it just the biomass value that characterizes the vegetation?
Is it assumed that land use categories have no biomass (and no change over time in biomass) if the vegetation has been cleared (this is answered on page 10)?

**Response:** Vegetation remaining in land-use categories 5-8 is indeed tracked. This vegetation has three pathways: 1) If its land-use type remains the same, its biomass will not grow as explained at the second paragraph of section 2.5, and 2) if it is converted to another type of 5-8, like from crop to managed pasture, its biomass will be cleared; 3) if it is converted back to secondary forest or non-forest, its biomass will continue growing, tracked by Eq.7.

Land-use categories 5-8 have biomass only if they have vegetation that came from primary/secondary land and was not cleared due to translation rule.

**Reviewer 1:** page 6, lines 11-14. Is this correct for gamma? It seems like it should be the opposite: a 1 value for "O" such that the land use type gains vegetation when no clearing occurs. Clearing means that no vegetation would be gained.

**Response:** We have modified Eq.2 and replaced $\gamma_{ij}$ by $(1-\gamma_{ij})$. In this way, gramma is still 1 for X and F indicating no vegetation could be gained in Eq.1, and gramma is still 0 for O indicating amount of $a_{ij}$ vegetation could be gained.

**Reviewer 1:** page 7, lines 3-4. This needs clarification, as gamma isn't the same as for the reverse of transitions to land use categories. For example, any transition to land use from primary or secondary would generate a loss of vegetation in primary or secondary land, regardless of clearing, which would mean that gamma is always one for eq. 5; the lost vegetation fraction would either be in a land use category, or it has been cleared.

**Response:** Yes, you are right. Any land-use transitions from primary or secondary would result in vegetation loss in primary and secondary. The lost vegetation could be remained in cropland, pasture or rangeland if translation rules indicate O, or be removed if the rules indicate X/F. We have corrected it by removing $\gamma_{ji}$ from Eq.5.

**Reviewer 1:** page 11, line 12. reference table 4

**Response:** Change made.

**Reviewer 1:** page 11, line 13. I suggest being specific here, as table 4 shows the results. You don't need the e.g. clause, and you should state that 5 of 8 countries have values within range for rules 1-3 and 4 out of 8 for rule 4 (if i counted correctly)

**Response:** Good point. We have changed the line to "*The forest cover estimates from Rules 1-4 are generally well within the range of diagnostics. For example, 6 of 8 countries have estimates within the range for Rule 1, 2, and 3, and 5 of 8 countries for Rule 4.*"

**Reviewer 1:** page 11, line 16. It isn't clear what you mean here by larger difference and what these differences are. I assume you mean differences between rules 1-3 and rule 4.

**Response:** The difference is between Rule1-3 and Rule 4. We have rephrased it as "*China and Brazil are the two countries where Rules 1-3 and Rule 4 have relatively larger difference between their estimates, the difference between Rule 1, 2, 3 and Rule 4 are 1.17 million and 1.08 million for China and Brazil respectively.*"

**Reviewer 1:** page 12, line 22. I am not sure that this metric evaluates the heterogeneity. I suggest something like "…best capture carbon density globally…)

**Response:** It is changed to "According to this comparison, Rules 1-3 best capture the carbon density globally (Figure 8)."

**Reviewer 1:** page 13, lines 10-11. I suggest rephrasing rangeland part, as currently it isn't clear what the rule does when establishing rangeland. Rather than switching to leaving vegetation, state that for rangeland the rule clears all vegetation only if source land is forest.

**Response:** Those lines have been changed.

**To Reviewer #3:**

**Reviewer 3:** I appreciate that the authors reconsidered the wording throughout the manuscript and added analysis, now presenting a more nuanced discussion of their approach to test different rules for the translation between LUH2 land-use and land-cover for ESMs. However, their main conclusion (=recommendation of rule 1) is not covered by the results of the analysis and sufficient justification for the exclusive recommendation of this rule is still missing. The framing of the manuscript still indicates the opposite, e.g. by the following statement in the abstract:

*'Examinations at global, country, and grid scales indicate that the recommended translation rule for CMIP6 models is 1) completely clear vegetation in land-use changes from primary and secondary land (including both forested and non-forested) to cropland, urban land, and managed pasture; 2) completely clear vegetation in land-use changes from primary forest and/or secondary forest to rangeland; 3) keep vegetation in land-use changes from primary non-forest and/or secondary non-forest to rangeland. This confirms the translation rules suggested earlier in the HYDE dataset underlying LUH2.'*

(1) The examinations across scales do not exclusively indicate rule 1 (instead rules 2 and 3 are equally likely), which the authors also state in the manuscript and in the reply to the reviewers.

(2) The examinations do not confirm the translation rule suggested by HYDE. Instead, the earlier suggestion from HYDE is used as (the main) justification to pick rule 1 instead of rules 2 or 3.

One way out would be to be very clear about the fact that rule 1 is only recommended to achieve consistent implementation in future simulations (i.e., it would require to be a major point in the discussion and also in the abstract) and this recommendation is NOT a result of the analyses in this manuscript (as these show that with the same arguments also recommendation of rules 2 and 3 could be justified).

In this context, the manuscript would also benefit from a more critical discussion about the downsides of a consistent 'translation rule' (which is not necessarily supported by available data). In my opinion, it is reasonable to aim at a standardized translation between LUH transitions and ESM land cover. But such a standardization always comes at the cost of omitting uncertainties, instead of actually reducing them. If, for example, the 'added uncertainty of 43 PgC in CMIP6' (as stated in the discussions) is avoided by implementing a consistent 'translation rule' this does not necessarily mean that the uncertainty is not there anymore; it might be just not depicted in the ESM results anymore. Only if the authors could show by their analysis that one rule performs significantly better than others, this would be an indication for actually 'reducing' uncertainties.

If the authors do not want to put more emphasis on the consistency aspect and/or highlight the limitations of their results (i.e., basically we do not know about the 'correct' rule), they would need to show with their analysis that rule 1 outperforms the other rules.

In sum, I think it is a useful study/analysis and worth to be published, but requires more nuance in the presentation of results, limitations, and derived conclusions.

**Response:** We have rephrased the abstract and major content of discussion section according to your suggestions and reworded statements about uncertainty. We do think standardization of land-use data and translation to land-cover is very important and beneficial to model simulations and evaluations in CMIP6, and this is the major point of this study. Therefore, this study discusses possible impacts of translation rule choices on land cover and aims to provide insights into LUH2 implementation for CMIP6 models. Our evaluations suggest Rule 2 gives closer estimates of vegetation carbon to diagnostics than Rule 1 and Rule 3. However, given uncertainties in vegetation carbon diagnostics, we think certainly differentiation of Rules 1, 2 and 3 is difficult in this study. We have discussed limitations at discussion section and revised the statement that a consistent translation rule could eliminate added uncertainty in LULCC emissions. Please see the point-by-point response below.

**Reviewer 3:** P3 L1-3 The authors did not address the comment on the (non-)existence of 'global transition rules'

**Response:** We do agree that a global rule may not exist, and implementation of such rule is very likely to oversimplify the translation between land-use changes and land-cover changes. We have stated in discussion section that global rules may result in errors in land-use translation and discussed the possibility of spatially or temporally varied rules.
[Figure]

**Reviewer 3:** P3 L5-7 What is the basis for this statement, if it's not supported by literature? Some previous analysis? I think without a reference it is a misleading statement.

**Response:** We have rephrased this paragraph.
[Figure]

**Reviewer 3:** P4 L1-5 I am sure there are suitable legend translations for ESA CCI land cover as well and it's one of the most up-to-date datasets, but indeed it's not a critical issue.

**Response:** It will be very interesting and valuable for future work to evaluate these rules with ESA CCI product.

**Reviewer 3:** P4 L28 While I see that it is difficult to link biomass density to tree density as the authors state, I think it would be worth to give an indication which one of the forest definitions in the literature (and also the ones in the reference maps used for comparison) is closest to this 2 kgC/m² definition. This definition has the potential to affect the results and deserves some attention.

**Response:** It is difficult to indicate which forest definition is closest to 2 kgC/m2, because only one definition is used to derive satellite-based tree-cover to forest cover, namely the 30% threshold. Detailed discussion of the threshold choices is beyond the scope of this study and it is well discussed in Sexton et al 2016 (already cited). Besides, we think comparisons in Fig.3, Fig.4

and Fig.S1 could suggest which reference map best matches the 2 kgC/m² definition. We also pointed out at the last paragraph of section 3.2 that our estimates are closer to GFC than others.

**Reviewer 3:** P5 L1 The intention to include rules 5-9 is still not clear. Although it might be useful for test/sensitivity runs (also for the ESM community), I think it doesn't make sense to include them if the main purpose of the manuscript is to derive a realistic/recommended translation rule (where these rules by definition are not useful). In the results (incl. tables and figures) they are hardly revisited and rather add confusion to some of the results. In my opinion, the authors should decide to either include all rules in all results/tables/figures or stick to rules 1-4. To concentrate on a different set of rules at different sections of the results is confusing.

**Response:** As we emphasized at the last paragraph of section 2.2, inclusion of Rules 5-9 could be used to infer individual contribution to land cover change from cropland, pasture and so on, and inclusion of them does not mean they are realistic to be implemented. We still think inclusion of Rules 5-9 will be helpful to answer questions like what likely impacts on forest/carbon from Rules 5-9 are implemented and why we recommend not to use these Rules. We also have added forest cover and carbon density maps of Rules 5-9 to the figure. S6 and S7 are for completeness.

**Reviewer 3:** P10 L9-11 I see the authors intention to include the whole range of currently available forest reference maps. However, it would be still useful to give an indication which one is closest to the GLM2 forest definition. If we would know, for example, that one of the products has a similar forest definition, this could increase the confidence/justification for one of the rules.

**Response:** We have added such an indication in terms of spatial pattern at the last paragraph of section 3.2. The reason why to include multiple maps as reference is there is no such a map that undoubtedly has the closest definition with the GLM2. First, all reference maps define forest based on tree cover rather than the GLM2 uses biomass. Second, Figure 2 and 4 suggest different closest maps. The GLC2000 has the smallest difference from the GLM2 in terms of global forest area in Figure 2, but the GFC gives the smallest AAD in Figure 4. Besides, the evaluation of rules is not affected without indication of such reference map. For example, Figure 4 shows Rule 1, 2, and 3 consistently produce the smallest overall difference among Rule 4 and other rules regardless of which satellite-based forest cover is chosen as the reference.

**Reviewer 3:** P1 L30 Reference biomass is also close for rule 2 and 3.

**Response**: The abstract has been rephrased.

**Reviewer 3:** P1 L30 Should it be: '[…] regions with forest cover larger than 50%'?

**Response:** The abstract has been rephrased.

**Reviewer 3:** P2 L16 As there is now already a carbon budget update, it might be good to use the latest values/reference. Friedlingstein, P. et al. 2019. Global Carbon Budget 2019. Earth Syst. Sci. Data 11, 1783–1838. https://doi.org/10.5194/essd-11-1783-2019

**Response:** Change made.

**Reviewer 3:** P3 L7-9 It is not only the lack of a globally consistent rule, but also the fact that the existence of such a global rule is very unlikely and a large simplification (see original comment P3 L1- 3).

**Response:** This paragraph has been rephrased in a way of emphasizing importance of consistent rules across models and standardization of LULCC data. We do agree that a global rule is very likely to over simplify the translation between land-use changes and land-cover changes, and we also think particular areas may need different rules. Therefore, we have discussed the possibility of spatially or temporally varied rules and noted readers the simplified rules designed in this study could result in errors at the third paragraph of section 4.

**Reviewer 3:** P3 L21-23 But also obscures the uncertainty from the lack of process understanding and lack of dedicated spatially explicit treatment.

**Response:** These lines have been changed as "*Therefore, a consistent rule across models for the LUH2 translation is needed with potential to reduce impacts of LUH2 use inconsistency on studying land-use effects through CMIP6*".

**Reviewer 3:** P3 L25 'which are then integrated'

**Response:** Change made.

**Reviewer 3:** P9 L20 'accounted for in bookkeeping model based studies'

**Response:** Change made.

**Reviewer 3:** P9 L29 'should be close to diagnostics'

**Response:** Change made.

**Reviewer 3:** P9 L31-33 It's not 'other criteria, such as …', but the only one that is used in the end to identify the recommended rule.

**Response**: Changed as **"**Finally, if several rules have a reasonably good fit to these three diagnostics, other criterion, namely the definition characteristics for managed …**"**

**Reviewer 3:** P10 L17-19 I don't understand what the authors intend to say here?

**Response:** Removed to avoid confusing.

**Reviewer 3:** P11 L2-4 Due to these large discrepancies it would be even more helpful to guide the reader with some information about which forest definition (of the reference maps) is closest to the GLM forest definition. (see original comments P4 L28; P10 L9-11).

**Response:** We have made such indication at the last paragraph of section 3.2.

**Reviewer 3:** P11 L17-18 And are within the range for Brazil, US, Congo, Indonesia, Peru.

**Response:** Added "are within range for Brazil, Democratic Republic of the Congo, Indonesia, and Peru". Rule 7 is outside the range for US.

**Reviewer 3:** P12 L6-14 All the realistic rules (1-4) reduce the pasture anomaly. Is this then just the difference between LUH1 and LUH2 or really a characteristic of the individual rules?

**Response:** Improvement of LUH2 itself primarily reduces the anomalous emissions by 6 Pg C, and choice of some rules could further reduce the emissions. We also clarified this by adding "*Rule 1 reduces anomalous emissions by 6 Pg C, indicating the sole contribution of the LUH2 to mitigate pasture anomaly*".

**Reviewer 3:** P13 L3-5 On average and globally. The regional and gridded comparisons (Table 4, Supplements) indicate that this might not hold at the country and grid level. Misleading statement.

**Response:** These lines have been changed.

**Reviewer 3:** P13 L6-8 It's actually hard to say if it is 'better' given all the uncertainties in these comparisons.

**Response:** Changed 'better' to 'closer'.

**Reviewer 3:** P13 L14-16 Which is also true for rule 2 and 3.

**Response:** This paragraph has been re-organized.

**Reviewer 3:** P13 L23-25 The uncertainty is not really reduced by implementing a consistent rule, as long as we do not know, which rule is 'correct'. It's just omitted from evaluation.

**Response:** This paragraph has been re-organized.

[revised manuscript text omitted]

**Figure S5. Carbon density difference comparison between the IPCC Tier-1 biomass density map and estimation of Rules 1-3. (a) Shaded regions represent where Rules 1-3 differ in estimates of carbon density; (b) Histogram of carbon density difference of shaded regions in (a), shared bounds present shift range of zero line under three assumed bias levels of the IPCC Tier-1 biomass. (c) – (f) are regional comparison of carbon density difference of Rules 1-3, regions where Rules 1-3 have the same estimate of carbon density are not shown.**

[Figure]

**Figure S6. Forest cover in 2000 from the Rules 5-9 respectively.**

[Figure]

**Figure S7. Global carbon density (above- and below-ground) maps estimated by Rules 5-9 respectively.**